# CFlowNets: Continuous Control with Generative Flow Networks

**Yinchuan Li**[1]**, Shuang Luo**[2][*]**, Haozhi Wang**[3]**, Jianye Hao**[1,3]
[1]Huawei Noah's Ark Lab, Beijing, China
[2]Zhejiang University, Huangzhou, China
[3]Tianjin University, Tianjin, China
`{liyinchuan, haojianye}@huawei.com, luoshuang@zju.edu.cn`
`wanghaozhi@tju.edu.cn`

## Abstract

Generative flow networks (GFlowNets), as an emerging technique, can be used as an alternative to reinforcement learning for exploratory control tasks. GFlowNet aims to generate distribution proportional to the rewards over terminating states, and to sample different candidates in an active learning fashion. GFlowNets need to form a DAG and compute the flow matching loss by traversing the inflows and outflows of each node in the trajectory. No experiments have yet concluded that GFlowNets can be used to handle continuous tasks. In this paper, we propose generative continuous flow networks (CFlowNets) that can be applied to continuous control tasks. First, we present the theoretical formulation of CFlowNets. Then, a training framework for CFlowNets is proposed, including the action selection process, the flow approximation algorithm, and the continuous flow matching loss function. Afterward, we theoretically prove the error bound of the flow approximation. The error decreases rapidly as the number of flow samples increases. Finally, experimental results on continuous control tasks demonstrate the performance advantages of CFlowNets compared to many reinforcement learning methods, especially regarding exploration ability.

## 1 Introduction

As an emerging technology, generative flow networks (GFlowNets) (Bengio et al., 2021a;b) can make up for the shortcomings of reinforcement learning (Kaelbling et al., 1996; Sutton & Barto, 2018) on exploratory tasks. Specifically, based on the Bellman equation (Sutton & Barto, 2018), reinforcement learning is usually trained to maximize the expectation of future rewards; hence the learned policy is more inclined to sample action sequences with higher rewards. In contrast, the training goal of GFlowNets is to define a distribution proportional to the rewards over terminating states, i.e., the parent states of the final states, rather than generating a single high-reward action sequence (Bengio et al., 2021a). This is more like sampling different candidates in an active learning setting (Bengio et al., 2021b), thus better suited for exploration tasks.

GFlowNets construct the state transitions of trajectories into a directed acyclic graph (DAG) structure. Each node in the graph structure corresponds to a different state, and actions correspond to transitions between different states, that is, an edge connecting different nodes in the graph. For discrete tasks, the number of nodes in this graph structure is limited, and each edge can only correspond to one discrete action. However, in real environments, the state and action spaces are continuous for many tasks, such as quadrupedal locomotion (Kohl & Stone, 2004), autonomous driving (Kiran et al., 2021; Shalev-Shwartz et al., 2016; Pan et al., 2017), or dexterous in-hand manipulation (Andrychowicz et al., 2020). Moreover, the reward distributions corresponding to these environments may be multimodal, requiring more diversity exploration. The needs of these environments closely match the strengths of GFlowNets. (Bengio et al., 2021b) proposes an idea for adapting GFlowNets to continuous tasks by replacing sums with integrals for continuous variables,

---

[*]Corresponding Author: Shuang Luo. This work was completed while Shuang Luo and Haozhi Wang were members of the Huawei Noah's Ark Lab for advanced study.

and they suggest the use of integrable densities and detailed balance (DB) or trajectory balance (TB) Malkin et al. (2022) criterion to obtain tractable training objectives, which can avoid some integration operations. However, this idea has not been verified experimentally.

In this paper, we propose generative Continuous Flow Networks, named CFlowNets for short, for continuous control tasks to generate policies that can be proportional to continuous reward functions. Applying GFlowNets to continuous control tasks is exceptionally challenging. In generative flow networks, the transition probability is defined as the ratio of action flow and state flow. For discrete state and action spaces, we can form a DAG and compute the state flow by traversing a node's incoming and outgoing flows. Conversely, it is impossible for continuous tasks to traverse all state-action pairs and corresponding rewards. To address this issue, we use important sampling to approximate the integrals over inflows and outflows in the flow-matching constraint, where we use a deep neural network to predict the parent nodes of each state in the sampled trajectory. The main contributions of this paper are summarized as the following:

**Main Contributions:** 1) We extend the theoretical formulation and flow matching theorem of previous GFlowNets to continuous scenarios. Based on this, a loss function for training CFlowNets is presented; 2) We propose an efficient way to sample actions with probabilities approximately proportional to the output of the flow network, and propose a flow sampling approach to approximate continuous inflows and outflows, which allows us to construct a continuous flow matching loss; 3) We theoretically analyze the error bound between sampled flows and inflows/outflows, and the tail becomes minor as the number of flow samples increases; 4) We conduct experiments based on continuous control tasks to demonstrate that CFlowNets can outperform current state-of-the-art RL algorithms, especially in terms of exploration capabilities. To the best of our knowledge, our work is the first to empirically demonstrate the effectiveness of flow networks on continuous control tasks. The codes are available at http://gitee.com/mindspore/models/tree/master/research/gflownets/cflownets

## 2 PRELIMINARIES

### 2.1 MARKOV DECISION PROCESS

A stochastic, discrete-time and sequential decision task can be described as a Markov Decision Process (MDP) , which is canonically formulated by the tuple:

$$M = \langle \mathcal{S}, \mathcal{A}, P, R, \gamma \rangle. \tag{1}$$

In the process, $s \in \mathcal{S}$ represents the state space of the environment. At each time step, agent receives a state $s$ and selects an action $a$ on the action space $\mathcal{A}$. This results in a transition to the next state $s'$ according to the state transition function $P(s'|s, a) : \mathcal{S} \times \mathcal{A} \times \mathcal{S} \rightarrow [0, 1]$. Then the agent gets the reward $r$ based on the reward function $R(s, a) : \mathcal{S} \times \mathcal{A} \rightarrow \mathbb{R}$. A stochastic policy $\pi$ maps each state to a distribution over actions $\pi(\cdot|s)$ and gives the probability $\pi(a|s)$ of choosing action $a$ in state $s$. The agent interacts with the environment by executing the policy $\pi$ and obtaining the admissible trajectories $\{(s_t, a_t, r_t, s_{t+1})\}_{t=1}^{n}$, where $n$ is the trajectory length. The goal of an agent is to maximize the discounted return $\mathbb{E}_{s_{0:n}, a_{0:n}} \left[ \sum_{t=0}^{\infty} \gamma^t r_t \mid s_0 = s, a_0 = a, \pi \right]$, where $\mathbb{E}$ is the expectation over the distribution of the trajectories and $\gamma \in [0, 1)$ is the discount factor.

### 2.2 GENERATIVE FLOW NETWORK

GFlowNet sees the MDP as a flow network. Define $s' = T(s, a)$ and $F(s)$ as the node's transition and the total flow going through $s$. Define an edge/action flow $F(s, a) = F(s \rightarrow s')$ as the flow through an edge $s \rightarrow s'$. The training process of vanilla GFlowNets needs to sum the flow of parents and children through nodes (states), which depends on the discrete state space and discrete action space. The framework is optimized by the following flow consistency equations:

$$\sum_{s,a:T(s,a)=s'} F(s,a) = R(s') + \sum_{a' \in \mathcal{A}(s')} F(s', a'), \tag{2}$$

which means that for any node $s$, the incoming flow equals the outgoing flow, which is the total flow $F(s)$ of node $s$.

## 3    CFLOWNETS: THEORETICAL FORMULATION

Considering a continuous task with tuple $(\mathcal{S}, \mathcal{A})$, where $\mathcal{S}$ denotes the continuous state space and $\mathcal{A}$ denotes the continuous action space. Define a trajectory $\tau = (s_1, ..., s_n)$ in this continuous task as a sequence sampled elements of $\mathcal{S}$ such that every transition $a_t : s_t \rightarrow s_{t+1} \in \mathcal{A}$. Further, we define an acyclic trajectory $\tau = (s_1, ..., s_n)$ as a trajectory satisfies the acyclic constraint: $\forall s_m \in \tau, s_k \in \tau, m \neq k$, we have $s_m \neq s_k$. Denote $s_0$ and $s_f$ respectively as the initial state and the final state related with the continuous task $(\mathcal{S}, \mathcal{A})$, we define the complete trajectory as any sampled acyclic trajectory from $(\mathcal{S}, \mathcal{A})$ starting in $s_0$ and ending in $s_f$. Correspondingly, a transition $s \rightarrow s_f$ into the final state is defined as the terminating transition, and $F(s \rightarrow s_f)$ is a terminating flow.

A trajectory flow $F(\tau) : \tau \mapsto \mathbb{R}^+$ is defined as any nonnegative function defined on the set of complete trajectories $\tau$. For each trajectory $\tau$, the associated flow $F(\tau)$ contains the number of *particles* (Bengio et al., 2021b) sharing the same path $\tau$. In addition, the tuple $(\mathcal{S}, \mathcal{A}, F)$ is called a continuous flow network. Let $T(s, a) = s'$ indicate an action $a$ that could make a transition from state $s$ to attain $s'$. Then we make the following assumptions.

**Assumption 1.** *Assume that the continuous take $(\mathcal{S}, \mathcal{A})$ is an "acyclic" task, which means that arbitrarily sampled trajectories $\tau$ are acyclic, i.e.,*

$$s_i \neq s_j, \forall s_i, s_j \in \tau = (s_0, ..., s_n), i \neq j.$$

**Assumption 2.** *Assume the flow function $F(s, a)$ is Lipschitz continuous, i.e.,*

$$|F(s, a) - F(s, a')| \leq L||a - a'||, \ a, a' \in \mathcal{A}, \tag{3}$$

$$|F(s, a) - F(s', a)| \leq L||s - s'||, \ s, s' \in \mathcal{S}, \tag{4}$$

*where $L$ is a constant.*

**Assumption 3.** *Assume that for any state pair $(s_t, s_{t+1})$, there is a unique action $a_t$ such that $T(s_t, a_t) = s_{t+1}$, i.e., taking action $a_t$ in $s_t$ is the only way to get to $s_{t+1}$. Hence we can define $s_t := g(s_{t+1}, a_t)$, where $g(\cdot)$ is a transition function. And assume actions are the translation actions.*

The necessity and rationality of Assumptions 1-3 are analyzed in the appendix. Under Assumption 1, we define the parent set $\mathcal{P}(s_t)$ of a state $s_t$ as the set that contains all of the direct parents of $s_t$ that could make a direct transition to $s_t$, i.e., $\mathcal{P}(s_t) = \{s \in \mathcal{S} : T(s, a \in \mathcal{A}) = s_t\}$. Similarly, define the child set $\mathcal{C}(s_t)$ of a state $s_t$ as the set contains all of the direct children of $s_t$ that could make a direct transition from $s_t$, i.e., $\mathcal{C}(s_t) = \{s \in \mathcal{S} : T(s_t, a \in \mathcal{A}) = s\}$. Then, we have the following continuous flow definitions, where Assumptions 2-3 make these integrals integrable and meaningful.

**Definition 1** (Continuous State Flow). *The continuous state flow $F(s) : \mathcal{S} \mapsto \mathbb{R}$ is the integral of the complete trajectory flows passing through the state:*

$$F(s) = \int_{\tau : s \in \tau} F(\tau) \mathrm{d}\tau. \tag{5}$$

**Definition 2** (Continuous Inflows). *For any state $s_t$, its inflows are the integral of flows that can reach state $s_t$, i.e.,*

$$\int_{s \in \mathcal{P}(s_t)} F(s \rightarrow s_t) \mathrm{d}s = \int_{s : T(s, a) = s_t} F(s, a) \mathrm{d}s = F(s_t) = \int_{a : T(s, a) = s_t} F(s, a) \mathrm{d}a, \tag{6}$$

*where $a : s \rightarrow s_t$ and $s = g(s_t, a)$ since Assumption 3 holds.*

**Definition 3** (Continuous Outflows). *For any state $s_t$, the outflows are the integral of flows passing through state $s_t$ with all possible actions $a \in \mathcal{A}$, i.e.,*

$$\int_{s \in \mathcal{C}(s_t)} F(s_t \rightarrow s) \mathrm{d}s = F(s_t) = \int_{a \in \mathcal{A}} F(s_t, a) \mathrm{d}a. \tag{7}$$

Based on the above definitions, we can define the transition probability $P(s \rightarrow s'|s)$ of edge $s \rightarrow s'$ as a special case of conditional probability introduced in Bengio et al. (2021b). In particular, the forward transition probability is given by

$$P_F(s_{t+1}|s_t) := P(s_t \rightarrow s_{t+1}|s_t) = \frac{F(s_t \rightarrow s_{t+1})}{F(s_t)}. \tag{8}$$

Similarly, the backwards transition probability is given by

$$P_B(s_t|s_{t+1}) := P(s_t \to s_{t+1}|s_{t+1}) = \frac{F(s_t \to s_{t+1})}{F(s_{t+1})}. \tag{9}$$

For any trajectory sampled from a continuous task $(\mathcal{S}, \mathcal{A})$, we have

$$\forall \tau = (s_1, ..., s_n), P_F(\tau) := \prod_{t=1}^{n-1} P_F(s_{t+1}|s_t) \tag{10}$$

$$\forall \tau = (s_1, ..., s_n), P_B(\tau) := \prod_{t=1}^{n-1} P_B(s_t|s_{t+1}), \tag{11}$$

and we further have

$$\forall s \in \mathcal{S} \backslash \{s_f\}, \int_{s' \in \mathcal{C}(s)} P_F(s'|s)\mathrm{d}s' = 1 \text{ and } \forall s \in \mathcal{S} \backslash \{s_0\}, \int_{s' \in \mathcal{P}(s)} P_B(s'|s)\mathrm{d}s' = 1. \tag{12}$$

Given any trajectory $\tau = (s_0, ..., s_n, s)$ that starts in $s_0$ and ends in $s$, a Markovian flow (Bengio et al., 2021b) is defined as the flow that satisfies

$$P(s \to s'|\tau) = P(s \to s'|s) = P_F(s'|s),$$

and the corresponding flow network $(\mathcal{S}, \mathcal{A}, F)$ is called a Markovian flow network (Bengio et al., 2021b). Then, we present Theorem 1 proved in the appendix B.1, which is an extension of Proposition 19 in Bengio et al. (2021b) to continuous scenarios.

**Theorem 1** (Continuous Flow Matching Condition). *Consider a non-negative function $\hat{F}(s, a)$ taking a state $s \in \mathcal{S}$ and an action $a \in \mathcal{A}$ as inputs. Then we have $\hat{F}$ corresponds to a flow if and only if the following continuous flow matching conditions are satisfied:*

$$\forall s' > s_0, \ \hat{F}(s') = \int_{s \in \mathcal{P}(s')} \hat{F}(s \to s')\mathrm{d}s = \int_{s:T(s,a)=s'} \hat{F}(s, a : s \to s')\mathrm{d}s$$

$$\forall s' < s_f, \ \hat{F}(s') = \int_{s'' \in \mathcal{C}(s')} \hat{F}(s' \to s'')\mathrm{d}s'' = \int_{a \in \mathcal{A}} \hat{F}(s', a)\mathrm{d}a. \tag{13}$$

*Furthermore, $\hat{F}$ uniquely defines a Markovian flow $F$ matching $\hat{F}$ such that*

$$F(\tau) = \frac{\prod_{t=1}^{n+1} \hat{F}(s_{t-1} \to s_t)}{\prod_{t=1}^{n} \hat{F}(s_t)}. \tag{14}$$

Theorem 1 means that as long as any non-negative function satisfies the flow matching conditions, a unique flow is determined. Therefore, for sparse reward environments, i.e., $R(s) = 0, \ \forall s \neq s_f$, we can obtain the target flow by training a flow network that satisfies the flow matching conditions. Such learning machines are called CFlowNets, and we have the following continuous loss function:

$$\mathcal{L}(\tau) = \sum_{s_t=s_1}^{s_f} \left( \int_{s_{t-1} \in \mathcal{P}(s_t)} F(s_{t-1} \to s_t)\mathrm{d}s_{t-1} - R(s_t) - \int_{s_{t+1} \in \mathcal{C}(s_t)} F(s_t \to s_{t+1})\mathrm{d}s_{t+1} \right)^2.$$

However, obviously, the above continuous loss function cannot be directly applied in practice. Next, we propose a method to approximate the continuous loss function based on the sampled trajectories to obtain the flow model.

## 4 CFLOWNETS: TRAINING FRAMEWORK

For continuous tasks, it is usually difficult to access all state-action pairs to calculate continuous inflows and outflows. In the following, we propose the CFlowNets training framework to address this problem, which includes an action sampling process, a flow matching approximation process. Then, CFlowNets can be trained based on an approximate flow matching loss function.

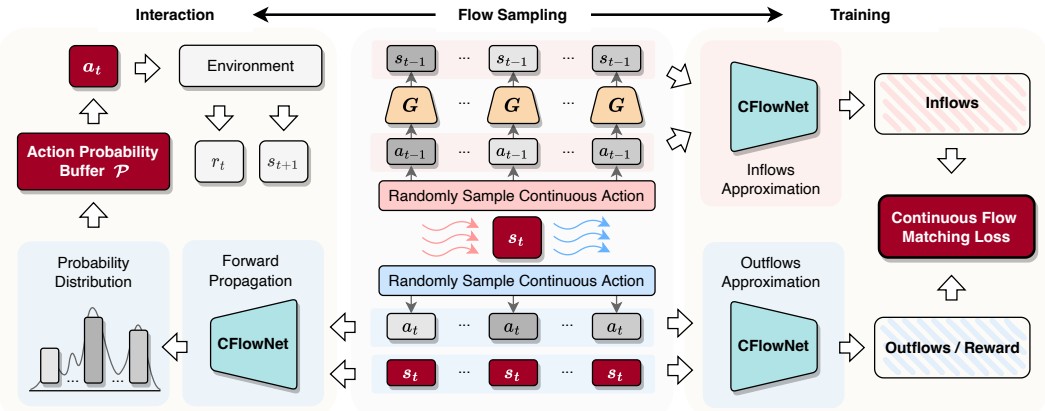

Figure 1: Overall framework of CFlowNets. **Left:** During the environment interaction phase, we sample actions to update states with probabilities proportional to the reward according to CFlowNet. **Middle:** We randomly sample actions to approximately calculate the inflows and outflows, where a DNN is used to estimate the parent states. **Right:** Continuous flow matching loss is used to train the CFlowNet based on making inflows equal to outflows or reward.

## 4.1 OVERALL FRAMEWORK

The overview framework of CFlowNets is shown in Figure 1, including the environment interaction, flow sampling, and training procedures. During the environment interaction phase (Left part of Figure 1), we sample an action probability buffer based on the forward-propagation of CFlowNets. We name this process the action selection procedure, as detailed in Section 4.2. After acquiring the action, the agent can interact with the environment to update the state, and this process repeats several steps until the complete trajectory is sampled. Once a buffer of complete trajectories is available, we randomly sample $K$ actions and compute the child states to approximately calculate the outflows. For the inflows, we use these sampled actions together with the current state as the input to the deep neural network $G$ to estimate the parent states. Based on these, we can approximately determine the inflows. We name this process the flow matching approximation procedure (Middle part of Figure 1), as detailed in Section 4.3. Finally, based on the approximate inflows and outflows, we can train a CFlowNet based on the continuous flow matching loss function (Right part of Figure 1), as details in Section 4.4. The pseudocode is provided in Appendix C.

## 4.2 ACTION SELECTION PROCEDURE

Starting from an empty set, CFlowNets aim to obtain complete trajectories $\tau = (s_0, s_1, ..., s_f) \in \mathcal{T}$ by iteratively sampling $a_t \sim \pi(a_t|s_t) = \frac{F(s_t, a_t)}{F(s_t)}$ with tuple $\{(s_t, a_t, r_t, s_{t+1})\}_{t=0}^{f}$. However, it is difficult to sample trajectories strictly according to the corresponding probability of $a_t$, since the actions are continuous, we cannot get the exact action probability distribution function based on the flow network $F(s_t, a_t)$. To solve this problem, at each state $s_t$, we first uniformly sample $M$ actions from $\mathcal{A}$ and generate an action probability buffer $\mathcal{P} = \{F(s_t, a_i)\}_{i=1}^{M}$, which is used as an approximation of action probability distributions. Then we sample an action from $\mathcal{P}$ according to the corresponding probabilities of all actions. Obviously, actions with larger $F(s_t, a_i)$ will be sampled with higher probability. In this way, we approximately sample actions from a continuous distribution according to their corresponding probabilities.

**Remark 1.** *After the training process, for tasks that require a larger reward, we can sample actions with the maximum flow output in $\mathcal{P}$ during the test process to obtain a relatively higher reward. How the output of the flow model is used is flexible, and we can adjust it for different tasks.*

## 4.3 FLOW MATCHING APPROXIMATION

Once a batch of trajectories $\mathcal{B}$ is available, to satisfy flow conditions, we require that for any node $s_t$, the inflows $\int_{a:T(s,a)=s_t} F(s, a) \mathrm{d}a$ equals the outflows $\int_{a \in \mathcal{A}} F(s_t, a) \mathrm{d}a$, which is the total flow

$F(s_t)$ of node $s_t$. However, obviously, we cannot directly calculate the continuous inflows and outflows to complete the flow matching condition. An intuitive idea is to discretize the inflows and outflows based on a reasonable approximation and match the discretized flows. To do this, we sample $K$ actions independently and uniformly from the continuous action space $\mathcal{A}$ and calculate corresponding $F(s_t, a_k), k = 1, ..., K$ as the outflows, i.e., we use the following approximation:

$$\int_{a \in \mathcal{A}} F(s_t, a)\mathrm{d}a \approx \frac{\mu(\mathcal{A})}{K} \sum_{k=1}^{K} F(s_t, a_k), \tag{15}$$

where $\mu(\mathcal{A})$ denotes the measure of the continuous action space $\mathcal{A}$.

By contrast, an approximation of inflow is more difficult since we should find the parent states first. To solve this problem, we construct a deep neural network $G$ (named "retrieval" neural network) parameterized by $\phi$ with $(s_{t+1}, a_t)$ as the input while $s_t$ as the output, and train this network based on $\mathcal{B}$ with the MSE loss. That is, we want use $G$ to fit function $g(\cdot)$. The network $G$ is usually easy to train since we consider tasks satisfy Assumption 3, and we can obtain a high-precision network $G$ through simple pre-training. As the training progresses, we can also occasionally update $G$ based on the sampled trajectories to ensure accuracy. Then, the inflows can be calculated approximately:

$$\int_{a:T(g(s_t, a), a) = s_t} F(g(s_t, a), a)\mathrm{d}a \approx \frac{\mu(\mathcal{A})}{K} \sum_{k=1}^{K} F(G_\phi(s_t, a_k), a_k). \tag{16}$$

Next, by assuming that the flow function $F(s, a)$ is Lipschitz continuous in Assumption 2, we could provide a non-asymptotic analysis for the error between the sample inflows/outflows and the true inflows/outflows. Theorem 2 establishes the error bound between the sample outflows (resp. inflows) and the actual outflows (resp. inflows) in the tail form and shows that the tail is decreasing exponentially. Furthermore, the tail gets much smaller with the increase of $K$, which means the sample outflows (resp. inflows) are a good estimation of the actual outflows (resp. inflows).

**Theorem 2.** *Let $\{a_k\}_{k=1}^{K}$ be sampled independently and uniformly from the continuous action space $\mathcal{A}$. Assume $G_{\phi^\star}$ can optimally output the actual state $s_t$ with $(s_{t+1}, a_t)$. For any bounded continuous action $a \in \mathcal{A}$ and any state $s_t \in \mathcal{S}$, we have*

$$\mathbb{P}\left(\left|\frac{\mu(\mathcal{A})}{K} \sum_{k=1}^{K} F(s_t, a_k) - \int_{a \in \mathcal{A}} F(s_t, a)\mathrm{d}a\right| \geq t\right) \leq 2\exp\left(-\frac{Kt^2}{2(L\mu(\mathcal{A})\mathrm{diam}(\mathcal{A}))^2}\right) \tag{17}$$

*and*

$$\mathbb{P}\left(\left|\frac{\mu(\mathcal{A})}{K} \sum_{k=1}^{K} F(G_{\phi^\star}(s_t, a_k), a_k) - \int_{a:T(s, a) = s_t} F(s, a)\mathrm{d}a\right| \geq t\right)$$
$$\leq 2\exp\left(-\frac{Kt^2}{2\big(L\mu(\mathcal{A})(\mathrm{diam}(\mathcal{A}) + \mathrm{diam}(\mathcal{S}))\big)^2}\right), \tag{18}$$

*where $L$ is the Lipschitz constant, $\mathrm{diam}(\mathcal{A})$ denotes the diameter of the action space $\mathcal{A}$ and $\mathrm{diam}(\mathcal{S})$ denotes the diameter of the state space $\mathcal{S}$.*

### 4.4 Loss Function

Based on (15) and (16), the continuous loss function can be approximated by

$$\mathcal{L}_\theta(\tau) = \sum_{s_t = s_1}^{s_f} \left[\sum_{k=1}^{K} F_\theta(G_\phi(s_t, a_k), a_k) - \lambda R(s_t) - \sum_{k=1}^{K} F_\theta(s_t, a_k)\right]^2, \tag{19}$$

where $\theta$ is the parameter of the flow network $F(\cdot)$ and $\lambda = K/\mu(\mathcal{A})$. Note that in many tasks we cannot obtain exact $\mu(\mathcal{A})$. For such tasks, we can directly set $\lambda$ to 1, and then adjust the reward shaping to ensure the convergence of the algorithm[1].

---

[1] A commonly used reward shaping method is to multiply the reward by a constant and adjust the reward to an appropriate range to ensure better convergence. Therefore, after setting $\lambda$ to 1, a reasonable reward shaping operation can also compensate for the influence of $\lambda$ error.

It is noteworthy that the magnitude of the state flow at different locations in the trajectory may not match. For example, the initial node flow is likely to be larger than the ending node flow. To solve this problem, inspired the log-scale loss introduced in GFlowNets (Bengio et al., 2021a), we can modify (19) into:

$$\mathcal{L}_\theta(\tau) = \sum_{s_t=s_1}^{s_f} \left\{ \log \left[ \epsilon + \sum_{k=1}^{K} \exp F_\theta^{\log}(G_\phi(s_t, a_k), a_k) \right] \right.$$
$$\left. - \log \left[ \epsilon + \lambda R(s_t) + \sum_{k=1}^{K} \exp F_\theta^{\log}(s_t, a_k) \right] \right\}^2, \qquad (20)$$

where $\epsilon$ is a hyper-parameter that helps to trade off small versus large flows and helps avoid the numerical problem of taking the logarithm of tiny flows. Note that Theorem 2 cannot be used to guarantee the unbiasedness of (20) because $\log \mathbb{E}(x) \neq \mathbb{E} \log(x)$. But experiments show that this approximation works well.

## 5 RELATED WORKS

**Generative Flow Networks.** Generative flow networks are proposed to enhance exploration capabilities by generating a distribution proportional to the rewards over terminating states (Bengio et al., 2021b;a). Since the network only samples actions based on the distribution of the corresponding rewards, rather than focusing only on actions that maximize rewards such as reinforcement learning, it can perform well on tasks with more diverse reward distributions, and has been successfully applied to molecule generation (Bengio et al., 2021a; Malkin et al., 2022; Jain et al., 2022), discrete probabilistic modeling (Zhang et al., 2022b), structure learning (Deleu et al., 2022), causal discovery Li et al. (2022) and graph neural network Li et al. (2023). The connection between deep generative models and GFlowNets is discussed in Zhang et al. (2022a) through the lens of Markov trajectory learning. In Bengio et al. (2021b), an idea is proposed for adapting GFlowNets to continuous tasks by replacing sums with integrals for continuous variables. Malkin et al. (2022) and Bengio et al. (2021b) propose detailed balance (DB) and trajectory balance (TB) objectives, which use parametric forward and backward policies in the objective function. These new objective functions do not require evaluating the flow model on multiple parents of a state, which is more efficient, especially for high-dimensional environments. Malkin et al. (2022) and Bengio et al. (2021b) mentioned that these objective functions can also be used in continuous scenarios by replacing the policy likelihoods in the objective with probability densities. A possible disadvantage is that it is not easy to estimate $P_F$ and $P_B$ in a continuous environment, since the state space is much larger than in a discrete scenario, and a small error in modeling probability densities can greatly affect the final performance. How to combine DB and TB with CFlowNets will be a worthy future work.

**Continuous Reinforcement Learning.** Policy gradient algorithms are widely used for reinforcement learning problems with continuous action spaces. The deterministic policy gradient (DPG) (Silver et al., 2014) algorithm is an actor-critic (Grondman et al., 2012; Rosenstein et al., 2004) method that uses an estimate of the learned value $Q(s, a)$ to train a deterministic policy $\mu : \mathcal{S} \rightarrow \mathcal{A}$ parameterized by $\theta^\mu$. Compared with CFlowNets, the policy is updated by applying the chain rule to the expected return $J$ from the start distribution with respect to the policy parameters:

$$\nabla_{\theta^\mu} J \approx \mathbb{E}_\mathcal{D} \left[ \nabla_{\theta^\mu} Q \left( s, a \mid \theta^Q \right) \big|_{a=\mu(s\mid\theta^\mu)} \right]$$
$$= \mathbb{E}_\mathcal{D} \left[ \nabla_a Q \left( s, a \mid \theta^Q \right) \big|_{a=\mu(s_t)} \nabla_{\theta^\mu} \mu \left( s \mid \theta^\mu \right) \right], \qquad (21)$$

where $\mathcal{D}$ is the replay buffer. The policy aims to maximize the expectation of future rewards, which are estimated by $Q$-learning. In this setting, the trajectories generated by the policy may be relatively homogeneous. However, the training goal of CFlowNets is to define a distribution proportional to the rewards over terminating states, resulting in more diverse trajectories that are beneficial for exploring the environment.

Later, deep DPG (DDPG) (Lillicrap et al., 2015) improves DPG and has good sample efficiency but suffers from extreme brittleness and hyperparameter sensitivity. Therefore, it is difficult to extend

DDPG to complex, high-dimensional tasks. To improve DDPG, twin delayed DDPG (TD3) (Fujimoto et al., 2018) adopts an actor-critic framework and considers the interaction between value update and function approximation error and in the policy. There are also some policy gradient (Sutton et al., 1999; Kohl & Stone, 2004; Khadka & Tumer, 2018) based algorithms that can be adapted for continuous tasks, such as proximal policy optimization (PPO) (Schulman et al., 2017) algorithms, asynchronous advantage actor-critic (A3C) (Stooke & Abbeel, 2018), and importance weighted actor-learner architecture (IMPALA) (Espeholt et al., 2018). PPO has the benefits of trust region policy optimization (Schulman et al., 2015), enabling multiple batches of data to be updated together. Therefore, it is simpler to implement, more general, and has lower sample complexity. Recently, phasic policy gradient (PPG) (Cobbe et al., 2021) is proposed to decouple the training between policy and value function while keeping their feature sharing, and PPG optimizes each objective with an appropriate level of sample reuse to improve sample efficiency. Most of these improved policy gradient methods can be classified as aiming at maximizing reward, so none of them are better suited for exploration tasks than CFlowNets.

Furthermore, some maximum entropy (Pitis et al., 2020; Haarnoja et al., 2018a; Hazan et al., 2019; Yarats et al., 2021) based reinforcement learning algorithms can also be adapted for continuous tasks, such as soft actor-critic (SAC) (Haarnoja et al., 2018b). By maximizing the expected reward and entropy, the actor network of SAC can successfully complete tasks while acting as randomly as possible. The difference between CFlowNets and SAC is: 1) SAC selects actions by a Gaussian policy, which is less expressive than using a general unnormalized action p.d.f. $F(s, a)$; 2) In the general case, SAC learns to be proportional to the long-term return, which generates the trajectory distribution satisfying $p(\tau) \propto R(\tau)$ with $R(\tau)$ is the return of $\tau$. CFlowNets considers all possible trajectories that lead to a terminal state $s_f$, and learn the policy to generate $s_f$ with $p(s_f) \propto R(s_f)$.

## 6 EXPERIMENTS

To demonstrate the effectiveness of the proposed CFlowNets, we conduct experiments on several continuous control tasks with sparse rewards, including Point-Robot-Sparse, Reacher-Goal-Sparse, and Swimmer-Sparse. The visualization of these environments is shown in Figures 7, 8 and 9. Then we compare CFlowNets with a few state-of-the-art baseline RL algorithms, such as DDPG (Lillicrap et al., 2015), TD3 (Fujimoto et al., 2018), PPO (Schulman et al., 2017), and SAC (Haarnoja et al., 2018b). More implementation details are provided in Appendix D.

Figure 2 illustrates the distributions of learned policies for CFlowNets and RL algorithms. All curves are max-min normalized. The gray curve is the ground truth of reward distribution generated by the agent's different actions when it goes to coordinates $(7, 7)$, which indicates that the optimal action here is to go right or up. The red curve shows the flow network output of CFlowNets under different actions, indicating that CFlowNets have an excellent fitting ability to the reward. In contrast, other reinforcement learning algorithms have difficulty fitting the actual reward distribution well.

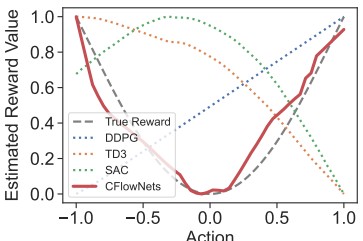

Figure 2: Reward distributions on Point-Robot-Sparse Task.

Figures 3(a)-(c) show the number of valid-distinctive trajectories explored as training progresses in Point-Robot-Sparse, Reacher-Goal-Sparse, and Swimmer-Sparse environment, respectively. After a certain number of training epochs, 10000 trajectories are collected. A valid-distinctive trajectory is defined as a reward above a threshold $\delta_r$ while the MSE between the trajectory and other trajectories is greater than another threshold $\delta_{\mathrm{mse}}$. That is, if the returns of both trajectories are high, but the two are close and the MSE is small, we consider it only one valid-distinctive exploration. $\delta_r$ in Point-Robot-Sparse, Reacher-Goal-Sparse, and Swimmer-Sparse is set as 0.5, -0.2, 5.0, respectively. $\delta_{\mathrm{mse}}$ in Point-Robot-Sparse, Reacher-Goal-Sparse, and Swimmer-Sparse is set as 0.02, 4.0, 1.0, respectively. As can be seen from the figure, DDPG, TD3 and PPO have the worst exploration ability, only one valid-distinctive trajectory is generated. SAC explores better at the beginning of training, and decreases as the training progresses and gradually converges. In contrast, the exploration ability of CFlowNets is very outstanding, the number of trajectories explored far exceeds other algorithms, and the exploration ability has been stable as the training progresses.

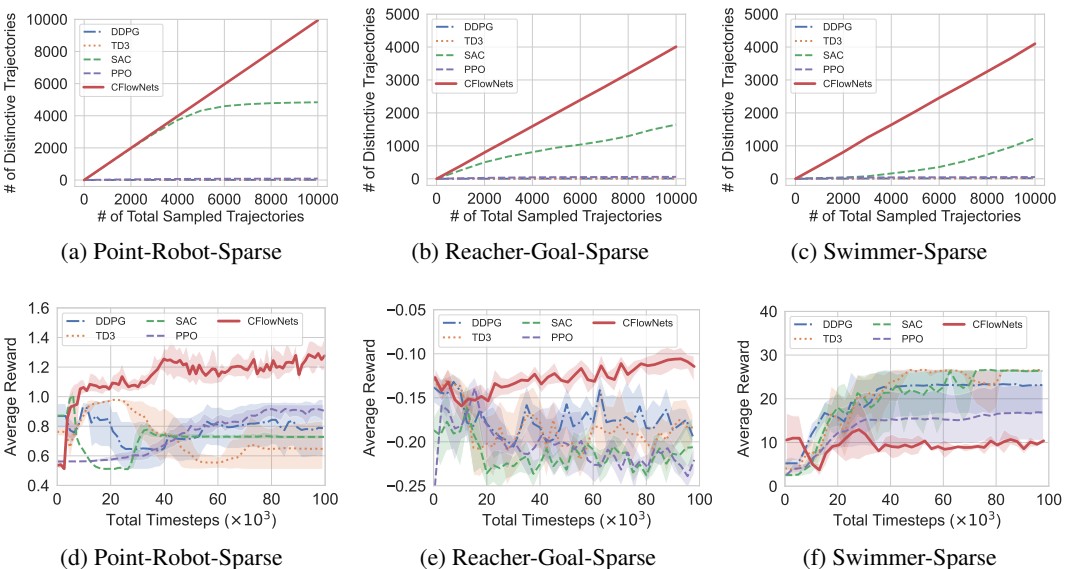

Figure 3: Comparison results of CFlowNets, DDPG, TD3, SAC and PPO on Point-Robot-Sparse, Reacher-Goal-Sparse, and Swimmer-Sparse tasks. **Top:** Number of valid-distinctive trajectories generated under 10000 explorations. **Bottom:** The average reward of different methods.

Figures 3(d)-(f) indicate the rewards during the training process in Point-Robot-Sparse, Reacher-Goal-Sparse, and Swimmer-Sparse environment, respectively. The shaded region represents 95% confidence interval across 5 runs. Figure 3(d) and Figure 3(e) show that CFlowNets has the fastest and more stable upward trend, and the final reward is ahead of that of other algorithms by a large margin. In contrast, CFlowNets do not perform as well as other algorithms in Figure 3(f). Since the rewards in Point-Robot-Sparse and Reacher-Goal-Sparse are more evenly distributed, so these two tasks are more inclined to exploration. CFlowNets has better exploration ability and hence can converge stably. As for Swimmer-Sparse, its reward distribution is relatively steep, and sampling near the maximum reward can achieve faster convergence. It is reasonable for CFlowNets to perform worse than RL on this task in terms of reward. However, in this environment, CFN can still maintain a good exploration ability.

# 7 CONCLUSION

In this paper, we propose generative continuous flow networks to enhance exploration in continuous control tasks. The theoretical formulation of CFlowNets is first presented. Then, a training framework for CFlowNets is proposed, including the action selection process, the flow approximation algorithm, and the continuous flow matching loss function. Theoretical analysis shows that the error of the flow approximation decreases rapidly as the number of flow samples increases. Experimental results on continuous control tasks illustrate the performance advantages of CFlowNets compared to many reinforcement learning methods. Especially in the exploration ability, the effect of CFlowNets far exceeds other state-of-the-art reinforcement learning algorithms.

**Limitations:** Similar to GFlowNets, CFlowNets aims to sample actions according to the flow network, rather than selecting actions with maximizing rewards. Therefore, CFlowNets are more suitable for exploration-biased tasks. It does not perform as well as reinforcement learning on tasks that aim to maximize reward. Of course, the purpose of CFlowNets is not to completely replace reinforcement learning, but as a supplement to reinforcement learning, giving a new option for continuous control tasks. **Future work:** Future work will be how to combine CFlowNets with DB and TB objective functions to improve training efficiency.

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

## A    DISCUSSIONS

### A.1    WHY IS ASSUMPTION 1 NECESSARY AND REASONABLE?

**Necessity:** For most environments, it is difficult to generate cycles when sampling a trajectory in a continuous space. Since $\forall t$, $\mu(\{s_0, ..., s_t\}) = 0$ and $\mu(\mathcal{A}) = \mu(\mathcal{A} \backslash \{s_0, ..., s_t\})$, that is, the probability of $s_{t+1} \in \{s_0, ..., s_t\}$ is very small. However, cycles often arise when certain environments have some special constraints. For example, a simple pendulum task (see Figure 4), the action is to control the pendulum to rotate from the previous position to the next position at a certain angle. For this task, it is difficult for a pendulum to rotate to exactly the same position in continuous space. However, if a wall is added to the task, the pendulum can easily go to the same position (see Figure 5), i.e. a cycle will occur. Therefore, we still need to add an acyclic assumption to make the theory and performance of CFlowNets guaranteed.

**Rationality:** This assumption is reasonable because for many continuous environments it is difficult to form cycles in trajectories without special constraints. Even for tasks prone to form cycles, we can directly add time steps in the state space to satisfy this assumption.

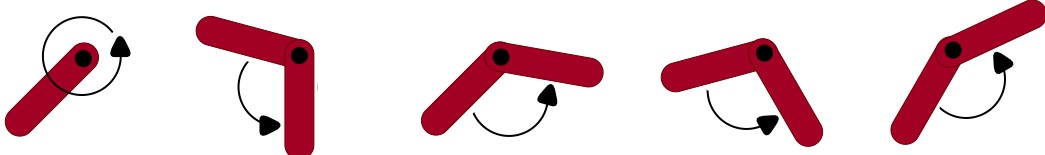

Figure 4: Pendulum. It is difficult for the state to be completely consistent in this continuous space.

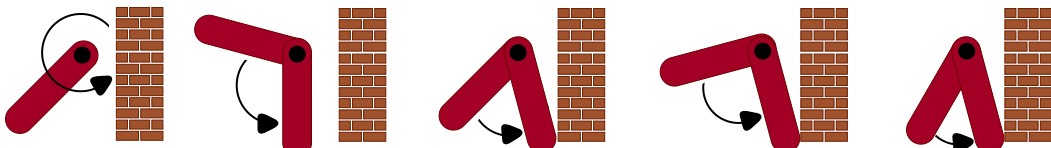

Figure 5: Pendulum-with-Wall. The state becomes consistent when reaching the wall.

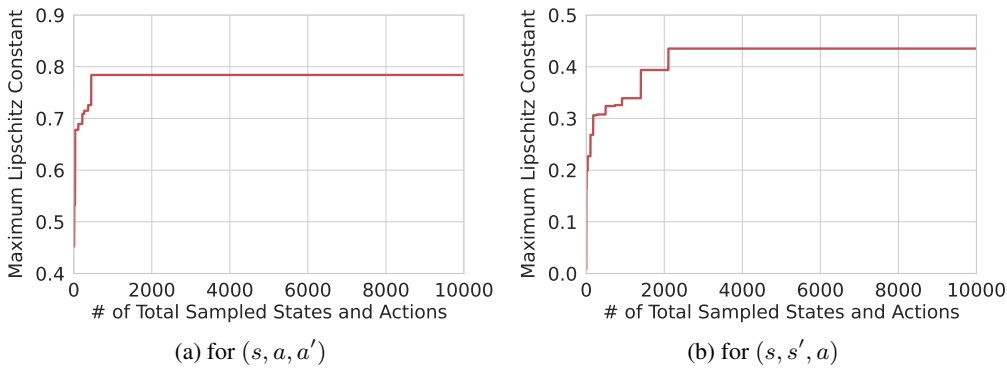

(a) for $(s, a, a')$          (b) for $(s, s', a)$

Figure 6: Accumulated maximum Lipschitz constant of flow network $F(s, a)$.

### A.2    WHY IS ASSUMPTION 2 NECESSARY AND REASONABLE?

**Necessity:** This assumption is mainly used to guarantee the existence of flow-related integrals, and to ensure that Theorem 2 holds.

**Rationality:** We justify this assumption based on simulations. As shown in Figure 6, we calculate $\frac{|F(s,a) - F(s,a')|}{\|a - a'\|}$ and $\frac{|F(s,a) - F(s',a)|}{\|s - s'\|}$ of each sample tuple $(s, a, a')$ and $(s, s', a)$ to analysis the Lipschitz constant, respectively. Their accumulated maximum Lipschitz constants are shown in Figures 6 (a) and (b), respectively. Clearly, there exists a finite Lipschitz constant for our flow network.

In addition, Lipschitz continuous is a common assumption of neural networks, just some quick examples: Du et al. (2019); Jacot et al. (2018); Allen-Zhu et al. (2019); Alistarh et al. (2018) all use this assumption to prove the convergence of algorithms.

### A.3 WHY IS ASSUMPTION 3 NECESSARY AND REASONABLE?

**Necessity:** This assumption is used in Definition 2 and enables the retrieval neural network to fit the function $g(s, a)$. While there is a one-to-one correspondence between most environment state transitions and actions, there are still some special cases where, given a state pair $(s, s')$, there can be an infinite number of actions. For example: for Pendulum-with-Wall in Figure 5, after reaching the wall, continuing to increase the action will not continue to change the state $s'$. In addition, a special case of the translation action could be $T(s, a) = s + a$ or using the special linear group, such that Definition 2 and 3 hold. The translation action is used to ensure that there is no Jacobian term in the continuous flow definition.

**Rationality:** This assumption is a property of many environments and therefore reasonable. For environments that do not satisfy this assumption, we can try to satisfy this assumption by modifying the state to add more information. For example, we can add the duration of the action to the state space of Pendulum-with-Wall task in Figure 5. Even if the action increases after reaching the wall, the position information will not be changed, but the duration will increase, so that the state transition and the action will correspond one-to-one. The worst case is that we cannot change the environment to satisfy the assumption. At this time, we mainly need to solve the problem that the output of the retrieval neural network $G$ cannot be multiple when the input is fixed. One of our conjectures is that maybe we can alleviate this problem by adding some small random noise to the input, but this idea has not been tested.

## B PROOFS

### B.1 PROOF OF THEOREM 1

**Theorem 1.** (Continuous Flow Matching Condition). *Consider a non-negative function $\hat{F}(s, a)$ taking a state $s \in \mathcal{S}$ and an action $a \in \mathcal{A}$ as inputs. Then we have $\hat{F}$ corresponds to a flow if and only if the following continuous flow matching conditions are satisfied:*

$$
\begin{aligned}
\forall s' > s_0, \ \hat{F}(s') &= \int_{s \in \mathcal{P}(s')} \hat{F}(s \to s') \mathrm{d}s = \int_{s:T(s,a)=s'} \hat{F}(s, a : s \to s') \mathrm{d}s \\
\forall s' < s_f, \ \hat{F}(s') &= \int_{s'' \in \mathcal{C}(s')} \hat{F}(s' \to s'') \mathrm{d}s'' = \int_{a \in \mathcal{A}} \hat{F}(s', a) \mathrm{d}a.
\end{aligned}
\tag{22}
$$

*Furthermore, $\hat{F}$ uniquely defines a Markovian flow $F$ matching $\hat{F}$ such that*

$$
F(\tau) = \frac{\prod_{t=1}^{n+1} \hat{F}(s_{t-1} \to s_t)}{\prod_{t=1}^{n} \hat{F}(s_t)}.
\tag{23}
$$

*Proof.* The proof is an extension of that of Proposition 19 in Bengio et al. (2021b) to the continuous case. We first prove the necessity part of the proof. Given a flow network, for non-initial and non-final nodes on a trajectory, the set of complete trajectories passing through state $s'$ is the union of the sets of trajectories going through $s \to s'$ for all $s \in \mathcal{P}(s')$, and also is the union of the sets of trajectories going through $s' \to s''$ for all $s'' \in \mathcal{C}(s')$, i.e.,

$$
\{\tau \in \mathcal{T} : s' \in \tau\} = \bigcup_{s \in \mathcal{P}(s')} \{\tau \in \mathcal{T} : s \to s' \in \tau\} = \bigcup_{s'' \in \mathcal{C}(s')} \{\tau \in \mathcal{T} : s' \to s'' \in \tau\}.
$$

Then we have

$$
F(s') = \int_{\tau: s' \in \tau} F(\tau) \mathrm{d}\tau = \int_{s \in \mathcal{P}(s')} \int_{\tau: s \to s' \in \tau} F(\tau) \mathrm{d}\tau \mathrm{d}s = \int_{s \in \mathcal{P}(s')} F(s \to s') \mathrm{d}s,
$$

and

$$F(s') = \int_{\tau:s' \in \tau} F(\tau)\mathrm{d}\tau = \int_{s'' \in \mathcal{C}(s')} \int_{\tau:s' \to s'' \in \tau} F(\tau)\mathrm{d}\tau \mathrm{d}s'' = \int_{s'' \in \mathcal{C}(s')} F(s' \to s'')\mathrm{d}s''.$$

Then we finish the necessity part. Next we show sufficiency. Let $\hat{Z} = \hat{F}(s_0)$ be the partition function and $\hat{P}_F$ be the forward probability function, then there exists a unique Markovian flow $F$ with forward transition probability function $P_F = \hat{P}_F$ and partition function $Z$ according to Proposition 18 in Bengio et al. (2021b), and such that

$$F(\tau) = \hat{Z} \prod_{t=1}^{n+1} \hat{P}_F(s_t|s_{t-1}) = \frac{\prod_{t=1}^{n+1} \hat{F}(s_{t-1} \to s_t)}{\prod_{t=1}^{n} \hat{F}(s_t)}, \tag{24}$$

where $s_{n+1} = s_f$. In addition, according to Lemma 1, we have

$$\int_{\tau \in \mathcal{T}_{0,s}} P_B(\tau)\mathrm{d}\tau = \int_{\tau \in \mathcal{T}_{0,s}} \prod_{s_t \to s_{t+1} \in \tau} P_B(s_t|s_{t+1})\mathrm{d}\tau = 1.$$

**Lemma 1.** *Considering a continuous task $(\mathcal{S}, \mathcal{A})$, where we have the transition probabilities defined in (8) and (9). Define $\mathcal{T}_{s,f}$ and $\mathcal{T}_{0,s}$ as the set of trajectories sampled from a continuous task starting in $s$ and ending in $s_f$; and starting in $s_0$ and ending in $s$, respectively. Then we have*

$$\forall s \in \mathcal{S} \backslash \{s_f\}, \int_{\tau \in \mathcal{T}_{s,f}} P_F(\tau)\mathrm{d}\tau = 1 \tag{25}$$

$$\forall s \in \mathcal{S} \backslash \{s_0\}, \int_{\tau \in \mathcal{T}_{0,s}} P_B(\tau)\mathrm{d}\tau = 1. \tag{26}$$

Thus, we have for $s' \neq s_0$:

$$F(s') = \hat{Z} \int_{\tau \in \mathcal{T}_{0,s'}} \prod_{(s_t \to s_{t+1}) \in \tau} \hat{P}_F(s_{t+1}|s_t)\mathrm{d}\tau$$

$$= \hat{Z} \frac{\hat{F}(s')}{\hat{F}(s_0)} \int_{\tau \in \mathcal{T}_{0,s'}} \prod_{(s_t \to s_{t+1}) \in \tau} \hat{P}_B(s_t|s_{t+1})\mathrm{d}\tau = \hat{F}(s'). \tag{27}$$

Combine (27) with $P_F = \hat{P}_F$ yields $\forall s \to s' \in \mathcal{A}$, $F(s \to s') = \hat{F}(s \to s')$. Finally, according to Proposition 16 in Bengio et al. (2021b), for any Markovian flow $F'$ matching $\hat{F}$ on states and edges, we have $F'(\tau) = F(\tau)$, which shows the uniqueness property. Then we complete the proof. $\square$

### B.2 Proof of Theorem 2

**Theorem 2.** *Let $\{a_k\}_{k=1}^{K}$ be sampled independently and uniformly from the continuous action space $\mathcal{A}$. Assume $G_{\phi^\star}$ can optimally output the actual state $s_t$ with $(s_{t+1}, a_t)$. For any bounded continuous action $a \in \mathcal{A}$ and any state $s_t \in \mathcal{S}$, we have*

$$\mathbb{P}\left( \left| \frac{\mu(\mathcal{A})}{K} \sum_{k=1}^{K} F(s_t, a_k) - \int_{a \in \mathcal{A}} F(s_t, a)\mathrm{d}a \right| \geq t \right) \leq 2\exp\left( -\frac{Kt^2}{2(L\mu(\mathcal{A})\mathrm{diam}(\mathcal{A}))^2} \right) \tag{28}$$

*and*

$$\mathbb{P}\left( \left| \frac{\mu(\mathcal{A})}{K} \sum_{k=1}^{K} F(G_{\phi^\star}(s_t, a_k), a_k) - \int_{a:T(s,a)=s_t} F(s, a)\mathrm{d}a \right| \geq t \right)$$

$$\leq 2\exp\left( -\frac{Kt^2}{2\big(L\mu(\mathcal{A})(\mathrm{diam}(\mathcal{A}) + \mathrm{diam}(\mathcal{S}))\big)^2} \right), \tag{29}$$

*where $L$ is the Lipschitz constant, $\mathrm{diam}(\mathcal{A})$ denotes the diameter of the action space $\mathcal{A}$ and $\mathrm{diam}(\mathcal{S})$ denotes the diameter of the state space $\mathcal{S}$.*

*Proof.* First, we show that the expectation of sample outflow is the true outflow and the expectation of sample inflow is the true inflow in Lemma 2.

**Lemma 2.** *Let $\{a_k\}_{k=1}^K$ be sampled independently and uniformly from the continuous action space $\mathcal{A}$. Assume $G_{\phi^\star}$ can optimally output the actual state $s_t$ with $(s_{t+1}, a_t)$. Then for any state $s_t \in \mathcal{S}$, we have*

$$\mathbb{E}\left[\frac{\mu(\mathcal{A})}{K}\sum_{k=1}^K F(s_t, a_k)\right] = \int_{a\in\mathcal{A}} F(s_t, a)\mathrm{d}a \tag{30}$$

*and*

$$\mathbb{E}\left[\frac{\mu(\mathcal{A})}{K}\sum_{k=1}^K F(G_{\phi^\star}(s_t, a_k), a_k)\right] = \int_{a:T(s,a)=s_t} F(s, a)\mathrm{d}a, \tag{31}$$

*where $s = g(s_t, a)$.*

Then, define the following terms:

$$\Gamma_k = \frac{\mu(\mathcal{A})}{K}F(s_t, a_k) - \frac{1}{K}\int_{a\in\mathcal{A}} F(s_t, a)\mathrm{d}a = \frac{1}{K}\int_{a\in\mathcal{A}}[F(s_t, a_k) - F(s_t, a)]\,\mathrm{d}a \tag{32}$$

and

$$\Lambda_k = \frac{\mu(\mathcal{A})}{K}F(G_{\phi^\star}(s_t, a_k), a_k) - \frac{1}{K}\int_{a:T(s,a)=s_t} F(s, a)\mathrm{d}a \tag{33}$$

$$= \frac{1}{K}\int_{a:T(s,a)=s_t}[F(G_{\phi^\star}(s_t, a_k), a_k) - F(s, a)]\,\mathrm{d}a, \tag{34}$$

where $s = g(s_t, a)$.

Note that the variables $\{\Gamma_k\}_{k=1}^K$ are independent and $\mathbb{E}[\Gamma_k] = 0, k = 1, \ldots, K$ according to Lemma 2. So the following equations hold

$$\mathbb{P}\left(\left|\frac{\mu(\mathcal{A})}{K}\sum_{k=1}^K F(s_t, a_k) - \int_{a\in\mathcal{A}} F(s_t, a)\mathrm{d}a\right| \geq t\right) = \mathbb{P}\left(\left|\sum_{k=1}^K \Gamma_k\right| \geq t\right) \tag{35}$$

and

$$\mathbb{P}\left(\left|\frac{\mu(\mathcal{A})}{K}\sum_{k=1}^K F(G_{\phi^\star}(s_t, a_k), a_k) - \int_{a:T(s,a)=s_t} F(s, a)\mathrm{d}a\right| \geq t\right) = \mathbb{P}\left(\left|\sum_{k=1}^K \Lambda_k\right| \geq t\right). \tag{36}$$

Since $F(s, a)$ is a Lipschitz function, we have

$$|\Gamma_k| \leq \frac{1}{K}\int_{a\in\mathcal{A}}\left|F(s_t, a_k) - F(s_t, a)\right|\mathrm{d}a$$

$$\leq \frac{L}{K}\int_{a\in\mathcal{A}}||a_k - a||\mathrm{d}a \leq \frac{L\mu(\mathcal{A})\mathrm{diam}(\mathcal{A})}{K}. \tag{37}$$

Together with Assumption 3, that is, for any pair of $(s, a)$ satisfying $T(s, a) = s_t$, $a$ is unique if we fix $s$, we have

$$|\Lambda_k| \leq \frac{1}{K}\int_{a:T(s,a)=s_t}\left|F(G_{\phi^\star}(s_t, a_k), a_k) - F(s, a)\right|\mathrm{d}a$$

$$\leq \frac{1}{K}\int_{a:T(s,a)=s_t}\left|F(G_{\phi^\star}(s_t, a_k), a_k) - F(s, a_k) + F(s, a_k) - F(s, a)\right|\mathrm{d}a$$

$$\leq \frac{1}{K}\int_{a:T(s,a)=s_t} L||G_{\phi^\star}(s_t, a_k) - s|| + L||a_k - a||\mathrm{d}a$$

$$\leq \frac{L\mu(\mathcal{A})\big(\mathrm{diam}(\mathcal{A}) + \mathrm{diam}(\mathcal{S})\big)}{K}. \tag{38}$$

**Lemma 3** (Hoeffding's inequality, Vershynin (2018)). *Let $x_1, \ldots, x_K$ be independent random variables. Assume the variables $\{x_k\}_{k=1}^K$ are bounded in the interval $[T_l, T_r]$. Then for any $t > 0$, we have*

$$\mathbb{P}\left(\left|\sum_{k=1}^K (x_k - \mathbb{E}x_k)\right| \geq t\right) \leq 2\exp\left(-\frac{2t^2}{K(T_r - T_l)^2}\right). \tag{39}$$

Incorporating $T_r = \frac{L}{K}\mu(\mathcal{A})\mathrm{diam}(\mathcal{A})$ and $T_l = -\frac{L}{K}\mu(\mathcal{A})\mathrm{diam}(\mathcal{A})$ in Lemma 3 with (37), and incorporating $T_r = \frac{L}{K}\mu(\mathcal{A})(\mathrm{diam}(\mathcal{A}) + \mathrm{diam}(\mathcal{S}))$ and $T_l = -\frac{L}{K}\mu(\mathcal{A})(\mathrm{diam}(\mathcal{A}) + \mathrm{diam}(\mathcal{S}))$ in Lemma 3 with (38), we complete the proof. □

### B.3 PROOF OF LEMMA 1

**Lemma 1.** *Considering a continuous task $(\mathcal{S}, \mathcal{A})$, where we have the transition probabilities defined in (8) and (9). Define $\mathcal{T}_{s,f}$ and $\mathcal{T}_{0,s}$ as the set of trajectories sampled from a continuous task starting in $s$ and ending in $s_f$; and starting in $s_0$ and ending in $s$, respectively. Then we have*

$$\forall s \in \mathcal{S}\backslash\{s_f\}, \quad \int_{\tau \in \mathcal{T}_{s,f}} P_F(\tau)\mathrm{d}\tau = 1 \tag{40}$$

$$\forall s \in \mathcal{S}\backslash\{s_0\}, \quad \int_{\tau \in \mathcal{T}_{0,s}} P_B(\tau)\mathrm{d}\tau = 1. \tag{41}$$

*Proof.* We show by strong induction that (40) holds, mainly following the proof of Lemma 5 in Bengio et al. (2021b), and then extending to (41) is trivial. Define $d$ as the maximum trajectory length in $\mathcal{T}_{s,f}, s \neq s_f$, we have:

Base cases: If $d = 1$, then

$$\int_{\tau \in \mathcal{T}_{s,f}} P_F(\tau)\mathrm{d}\tau = P_F(s \to s_f) = 1$$

holds by noting $\mathcal{T}_{s,f} = \{(s \to s_f)\}$.

Induction steps: Consider $d > 1$, by noting (12) we have

$$\int_{\tau \in \mathcal{T}_{s,f}} P_F(\tau)\mathrm{d}\tau = \int_{s' \in \mathcal{C}(s)} \int_{\tau \in \mathcal{T}_{s \to s',f}} P_F(\tau)\mathrm{d}\tau\mathrm{d}s' \tag{42}$$

$$= \int_{s' \in \mathcal{C}(s)} \int_{\tau \in \mathcal{T}_{s',f}} P_F(s'|s)P_F(\tau)\mathrm{d}\tau\mathrm{d}s' \tag{43}$$

$$= \int_{s' \in \mathcal{C}(s)} P_F(s'|s)\mathrm{d}s' \int_{\tau \in \mathcal{T}_{s',f}} P_F(\tau)\mathrm{d}\tau = 1, \tag{44}$$

where the last equality follows by the induction hypotheses. □

### B.4 PROOF OF LEMMA 2

**Lemma 2.** *Let $\{a_k\}_{k=1}^K$ be sampled independently and uniformly from the continuous action space $\mathcal{A}$. Assume $G_{\phi^\star}$ can optimally output the actual state $s_t$ with $(s_{t+1}, a_t)$. Then for any state $s_t \in \mathcal{S}$, we have*

$$\mathbb{E}\left[\frac{\mu(\mathcal{A})}{K}\sum_{k=1}^K F(s_t, a_k)\right] = \int_{a \in \mathcal{A}} F(s_t, a)\mathrm{d}a \tag{45}$$

*and*

$$\mathbb{E}\left[\frac{\mu(\mathcal{A})}{K}\sum_{k=1}^K F(G_{\phi^\star}(s_t, a_k), a_k)\right] = \int_{a:T(s,a)=s_t} F(s, a)\mathrm{d}s, \tag{46}$$

*where $s = g(s_t, a)$.*

*Proof.* Since $\{a_k\}_{k=1}^K$ is sampled independently and uniformly from the continuous action space $\mathcal{A}$, then we have

$$\mathbb{E}\left[F(s_t, a_k)\right] = \frac{1}{\mu(\mathcal{A})} \int_{a \in \mathcal{A}} F(s_t, a)\mathrm{d}a. \tag{47}$$

Therefore, we obtain

$$\mathbb{E}\left[\frac{\mu(\mathcal{A})}{K} \sum_{k=1}^K F(s_t, a_k)\right] = \frac{\mu(\mathcal{A})}{K} \sum_{k=1}^K \mathbb{E}\left[F(s_t, a_k)\right] \tag{48}$$

$$= \int_{a \in \mathcal{A}} F(s_t, a)\mathrm{d}a. \tag{49}$$

Since Assumption 3 holds, for any pair of $(s, a)$ satisfying $T(s, a) = s_t$, $a$ is unique if we fix $s$, we have

$$\mathbb{E}\left[F(G_{\phi^\star}(s_t, a_k), a_k)\right] = \frac{1}{\mu(\mathcal{A})} \int_{a:T(s,a)=s_t} F(s, a)\mathrm{d}a,$$

where $s = g(s_t, a)$.

Therefore, we get

$$\mathbb{E}\left[\frac{\mu(\mathcal{A})}{K} \sum_{k=1}^K F(G_{\phi^\star}(s_t, a_k), a_k)\right] = \frac{\mu(\mathcal{A})}{K} \sum_{k=1}^K \mathbb{E}\left[F(G_{\phi^\star}(s_t, a_k), a_k)\right]$$

$$= \int_{a:T(s,a)=s_t} F(s, a)\mathrm{d}a.$$

Then we complete the proof. $\qquad\qquad\square$

## C    PSEUDOCODE OF CFLOWNETS

For clarity, we show pseudocode for CFlowNets in Algorithm 1.

---
**Algorithm 1** Generative Continuous Flow Networks (CFlowNets) Algorithm
---
**Initialize**: Flow network $\theta$; a pretrained retrieval network $G_\phi$; and empty buffer $\mathcal{D}$ and $\mathcal{P}$

1: **repeat**
2:    Set $t = 0$, $s = s_0$
3:    **while** $s \neq terminal$ and $t < T$ **do**
4:       Uniformly sample $M$ actions $\{a_i\}_{i=1}^M$ from action space $\mathcal{A}$
5:       Compute edge flow $F_\theta(s_t, a_i)$ for each $a_i \in \{a_i\}_{i=1}^M$ to generate $\mathcal{P}$
6:       Sample $a_t \sim \mathcal{P}$ and execute $a_t$ in the environment to obtain $r_{t+1}$ and $s_{t+1}$
7:       $t = t + 1$
8:    **end while**
9:    Store episodes $\{(s_t, a_t, r_t, s_{t+1})\}_{t=1}^T$ in replay buffer $\mathcal{D}$
10:   [Optional] Fine-tuning retrieval network $G_\phi$ based on $\mathcal{D}$
11:   Sample a random minibatch $\mathcal{B}$ of episodes from $\mathcal{D}$
12:   Uniformly sample $K$ actions $\{a_k\}_{k=1}^K$ from action space $\mathcal{A}$ for each state in $\mathcal{B}$
13:   Compute parent states according to $\{G_\phi(s, a_k)\}_{k=1}^K$ for each state in $\mathcal{B}$
14:   **Inflows:**
15:       $\log[\epsilon + \sum_{k=1}^K \exp F_\theta^{\log}(G_\phi(s_t, a_k), a_k)]$
16:   **Outflows or reward:**
17:       $\log[\epsilon + \lambda R(s_t) + \sum_{k=1}^K \exp F_\theta^{\log}(s_t, a_k)]$
18:   Update flow network $F_\theta$ according to (20)
19: **until** convergence
---

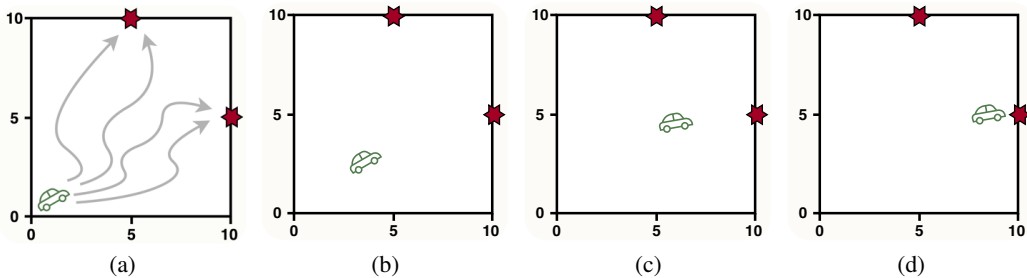

Figure 7: Visualization of Point-Robot-Sparse task.

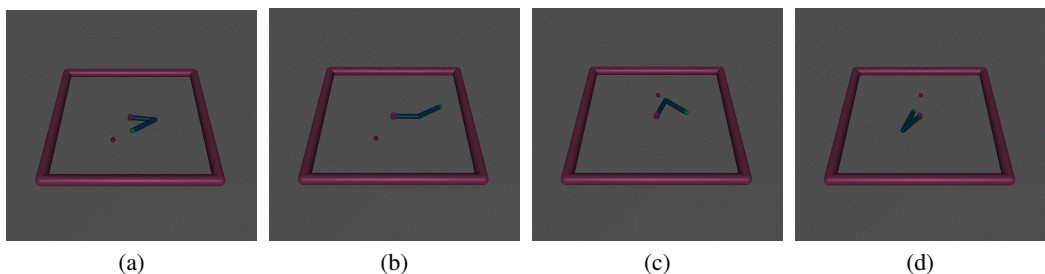

Figure 8: Visualization of Reacher-Goal-Sparse task.

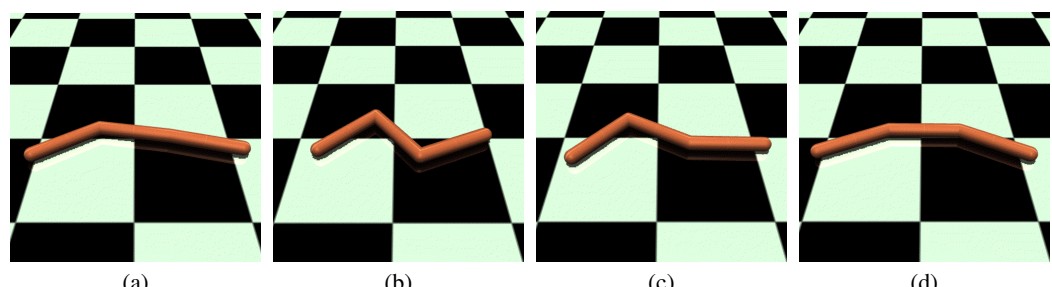

Figure 9: Visualization of Swimmer-Sparse task.

# D  ADDITIONAL EXPERIMENTS

## D.1  VISUALIZATION OF ENVIRONMENT

As shown in Figures 7, 8 and 9, we provide the visualization of Point-Robot-Sparse, Reacher-Goal-Sparse, and Swimmer-Sparse tasks. In Point-Robot-Sparse, the goal of the agent is to navigate two different goals. The agent starts at the starting coordinate $(0,0)$ and moves towards the target coordinate one step at a time. The environment has two target coordinates $(5,10)$ and $(10,5)$ with a maximum episode length of 12, and the environment returns a reward only when the last step is reached. Rewards are issued by measuring the distance between the agent's current position and the target node, and the closer the distance, the greater the reward. Each time the agent can take a step from any angle to the upper right.

Both Reacher-Goal-Sparse and Swimmer-Sparse are adapted from OpenAI Gym's MuJoCo environment. In the Reacher-Goal-Sparse, "Reacher" is a two-jointed robotic arm. The goal of the agent is to reach a randomly generated target by moving the robots end effector. Figure 8 shows the movement process of the robotic arm. By adjusting the torque applied at the hinge joint, the end effector can gradually approach the target. In the Swimmer-Spars, the "swimmer" is suspended in a

two-dimensional pool, and the goal is to move as fast as possible towards the right or left. Figure 9 shows the shape change process of the robot during motion. By taking the action that applies torque on the rotors and using the fluids friction, the robot can swim faster. We set the maximum number of steps to 50 for these two environments. For Reacher-Goal-Sparse, when the last step is reached, the environment returns a reward that measures how far the agent is from the randomly generated target. The closer the agent is to the target, the greater the reward. For Swimmer-Sparse, the farther to the left or right from the starting point, the greater the reward returned.

## D.2 ADDITIONAL ANALYSIS

Figure 10 shows that the average reward and reward distribution of different algorithms on the Point-Robot-OneGoal-Sparse task, where an agent needs to navigate to a specific location. Figure 10 (a) indicates that CFlowNets can obtain the highest average return compared to other RL-based algorithms. In Figure 10 (b), all algorithms are able to fit the reward distribution well under the one goal setting, while CFlowNets can achieve better. Note that RL algorithms can also learn the reward distribution in this task, since maximizing the reward is the optimal policy in the case of a single objective, and the policy is not difficult to learn.

In Figure 11, we provide the action reward distribution of different algorithms with 2e4 total timesteps on Point-Robot-Sparse with Point (4,8), Point (8,4) and Point (7,7), respectively. Note that unlike Figure 2, where the total number of timesteps is 1e5, here we show the result with 2e4 total timesteps since we found DDPG is overfit after 1e5 timesteps in this task. Therefore we show the results without overfitting for a fairer comparison. We can see that no matter at which point, the policy of CFlowNets can better match the real reward distribution. For example, at points (4,8) and (8,4), CFlowNet tends to choose actions that guide the agent towards (5, 10) and (10, 5), respectively. For a location between two goals (point (7,7)), there are two directions that allow the agent to reach goals with high rewards. In contrast, the policy learned by RL algorithms can only occasion-

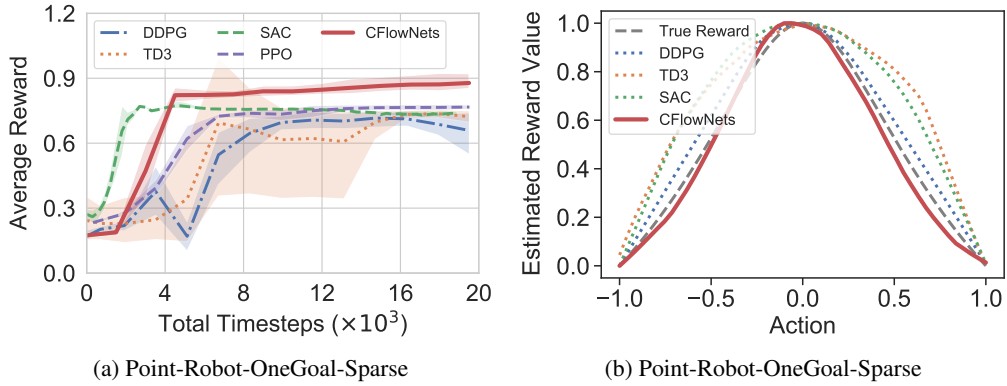

(a) Point-Robot-OneGoal-Sparse        (b) Point-Robot-OneGoal-Sparse

Figure 10: The average reward and reward distributions of CFlowNets, DDPG, TD3, SAC and PPO on Point-Robot-OneGoal-Sparse task.

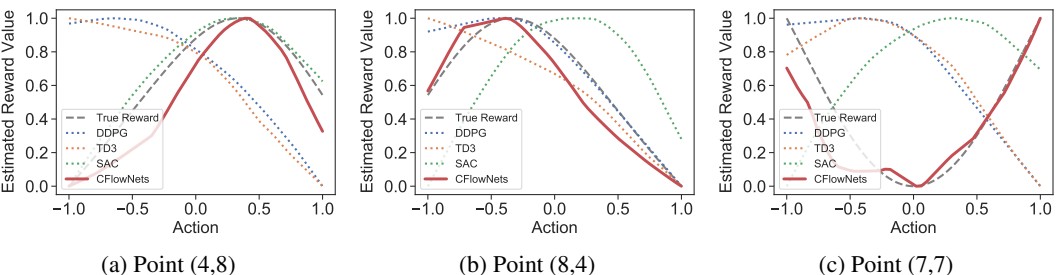

(a) Point (4,8)        (b) Point (8,4)        (c) Point (7,7)

Figure 11: The reward distributions of different points on Point-Robot-Sparse task.

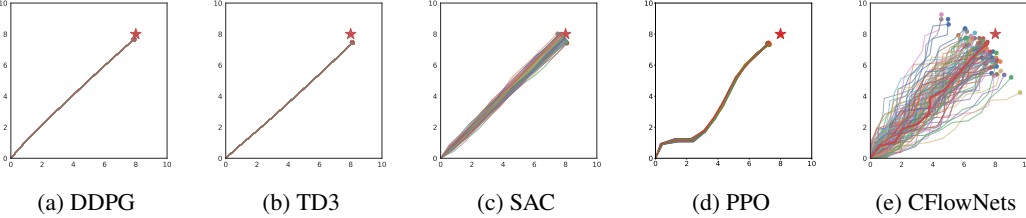

Figure 12: Sampled trajectories on Point-Robot-OneGoal-Sparse task.

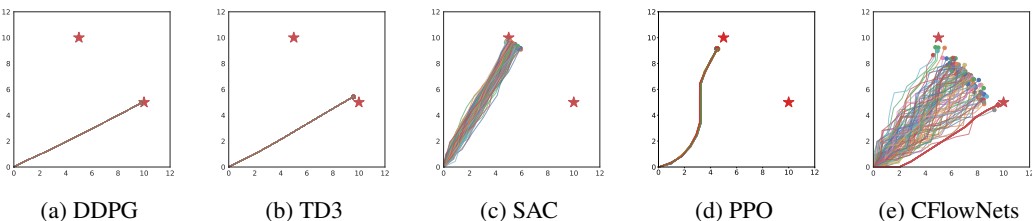

Figure 13: Sampled trajectories on Point-Robot-Sparse task.

ally match the true reward distribution of a certain point, and cannot stably match every point. This also shows that the policies learned by RL algorithms is relatively simple. CFlowNets learn more diverse policies for agents to reach different goals with high rewards, while other methods usually find one goal instead of all potentially high reward locations.

Figure 12 and Figure 13 show the results of trajectories visualization produced by different algorithms. In the Point-Robot-OneGoal-Sparse task, the trajectories of DDPG, TD3, and PPO are single, while SAC can select actions from the policy probability distribution, so different trajectories can be obtained. In contrast, CFlowNets found more diverse trajectories and also found the highest reward goal (thickened red trajectory), which means that CFlowNets can better explore the region near the goal. In the Point-Robot-Sparse task, the RL-based algorithms seek only one goal. However, CFlowNets can find all goals.

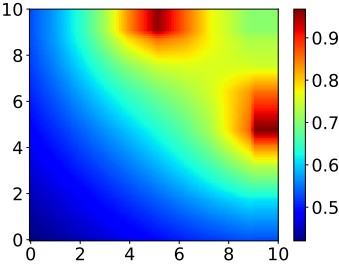

Figure 14: Reward distributions on Point-Robot-Sparse Task.

It is worth noting that in Figure 13 (e), the density of CFlowNets sampling trajectories is not as dense as in Figure 12 (e) near the maximum reward. Rather, it is denser on the diagonal. This is because in most positions, the action probability of choosing to go up and to the right is relatively high, so it is easier to go to the diagonal direction in combination. In addition, the reward on the line between two goals is not small. When sampling according to the output of the flow model as a probability, many trajectories themselves are more likely to reach the diagonal. Figure 14 shows the true reward distribution of Point-Robot-Sparse, where the reward is higher in the area near two goals and the line between two goals.

## D.3 EXPERIMENT RESULTS ON HIGHWAY-PARKING-SPARSE

We evaluate the performance of CFLowNets on Highway-Parking-Sparse, which is an ego-vehicle control task. As shown in Figure 15, the goal is to make the ego-vehicle park in a given space with the appropriate orientation by adjusting its controller. The dimension of the vehicle observation is 18, consisting of the distance between the vehicle and parking, the vehicle speed, the triangular heading information, the goal the agent should attempt to achieve, and the goal that it currently

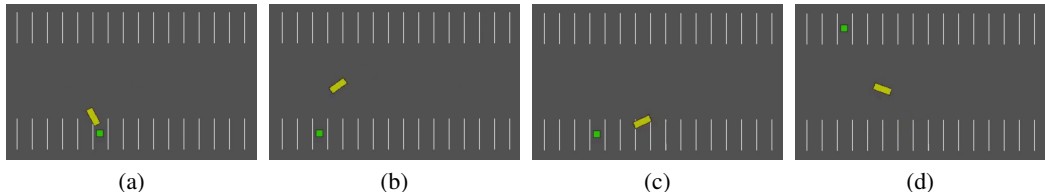

Figure 15: Visualization of Highway-Parking-Sparse task.

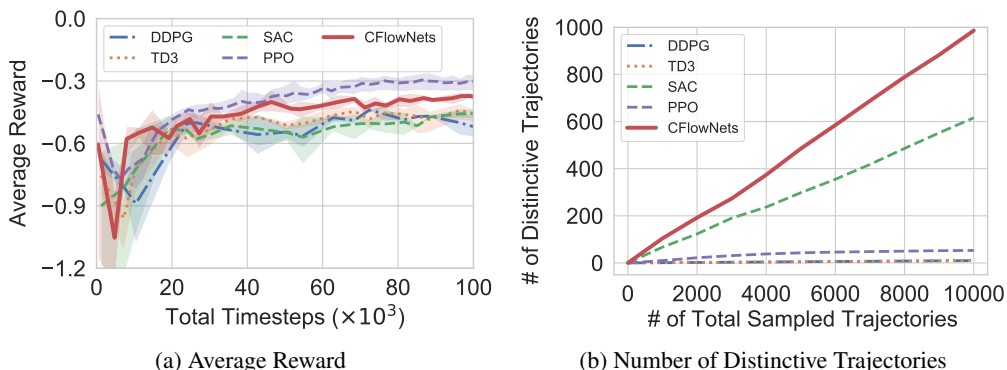

(a) Average Reward          (b) Number of Distinctive Trajectories

Figure 16: The average reward and number of valid-distinctive trajectories generated under 10000 explorations of CFlowNets, DDPG, TD3, and SAC on Highway-Parking-Sparse.

achieves. The action space includes control over the throttle and steering angle, and the reward function is set as the distance between the ego-vehicle and parking. Figure 16 shows the average reward and the number of valid-distinctive trajectories explored as training progresses of different algorithms, which illustrates that the performance of CFlowNets is more promising than other RL-based algorithms. Even for higher-dimensional continuous tasks, CFlowNets have very competitive reward results (outperforming DDPG, TD3, and SAC), while achieving much better exploration performance than RL-based algorithms.

## D.4 BASELINES

We compare our proposed CFlowNets to the following baselines:

- Deep Deterministic Policy Gradient (DDPG) (Lillicrap et al., 2015). `https://github.com/sfujim/TD3/blob/master/DDPG.py`
- Twin Delayed Deep Deterministic Policy Gradient (TD3) (Fujimoto et al., 2018). `https://github.com/sfujim/TD3`
- Soft Actor-Critic (SAC) (Haarnoja et al., 2018b). `https://github.com/denisyarats/pytorch_sac/`
- Proximal Policy Optimization (PPO) (Schulman et al., 2017). `https://github.com/DLR-RM/stable-baselines3/blob/master/stable_baselines3/ppo/ppo.py`

## D.5 HYPER-PARAMETER

We provide the hyper-parameters of all compared methods under different environments in Table 1, Table 2, Table 3, Table 4, and Table 5.

As for "Total Timesteps", "Start Traning Timestep", "Max Episode Length", "Actor Network Hidden Layers", "Critic Network Hidden Layers", "Optimizer", "Learning Rate", and "Discount Fac-

tor", we set them the same for all algorithms for a fair comparison. As for these specific parameters for baseline algorithms, we remain them the same as those in the original code to achieve good performance. As for these specific parameters of our CFlowNets, we set the number of sample flows to 100 and the action probability buffer size to 1000 to tradeoff the performance and computational load. Note that CFlowNets dose not require as large a replay buffer size as other RL algorithms, since the exploration ability of CFlowNets is better than that of others. And a good policy can already be learned from a small replay buffer. This is also an advantage of CFlowNets compared to RL based algorithms.

Table 1: Hyper-parameters of CFlowNets under different environments.

|  | Point-Robot-Sparse | Reacher-Goal-Sparse | Swimmer-Sparse |
| --- | --- | --- | --- |
| Total Timesteps | 100,000 | 100,000 | 100,000 |
| Start Traning Timestep | 4,000 | 7,500 | 7,500 |
| Max Episode Length | 12 | 50 | 50 |
| Flow Network Hidden Layers | [256,256] | [256,256] | [256,256] |
| Retrieval Network Hidden Layers | [256,256,256] | [256,256,256] | [256,256,256] |
| Optimizer | Adam | Adam | Adam |
| Learning Rate | 0.0003 | 0.0003 | 0.0003 |
| Batchsize | 128 | 128 | 128 |
| Number of Sample Flows | 100 | 100 | 100 |
| Action Probability Buffer Size | 1,000 | 1,0000 | 10,000 |
| Replay Buffer Size | 8,000 | 2,000 | 2,000 |
| $\epsilon$ | 1.0 | 1.0 | 1.0 |

Table 2: Hyper-parameter of DDPG under different environments.

|  | Point-Robot-Sparse | Reacher-Goal-Sparse | Swimmer-Sparse |
| --- | --- | --- | --- |
| Total Timesteps | 100,000 | 100,000 | 100,000 |
| Start Traning Timestep | 4,000 | 7,500 | 7,500 |
| Max Episode Length | 12 | 50 | 50 |
| Actor Network Hidden Layers | [256,256] | [256,256] | [256,256] |
| Critic Network Hidden Layers | [256,256] | [256,256] | [256,256] |
| Optimizer | Adam | Adam | Adam |
| Learning Rate | 0.0003 | 0.0003 | 0.0003 |
| Batchsize | 256 | 256 | 256 |
| Discount Factor | 0.99 | 0.99 | 0.99 |
| Replay Buffer Size | 100,000 | 100,000 | 100,000 |
| Target Network Update Rate | 0.005 | 0.005 | 0.005 |

Table 3: Hyper-parameter of TD3 under different environments.

|  | Point-Robot-Sparse | Reacher-Goal-Sparse | Swimmer-Sparse |
| --- | --- | --- | --- |
| Total Timesteps | 100,000 | 100,000 | 100,000 |
| Start Traning Timestep | 4,000 | 7,500 | 7,500 |
| Max Episode Length | 12 | 50 | 50 |
| Actor Network Hidden Layers | [256,256] | [256,256] | [256,256] |
| Critic Network Hidden Layers | [256,256] | [256,256] | [256,256] |
| Optimizer | Adam | Adam | Adam |
| Learning Rate | 0.0003 | 0.0003 | 0.0003 |
| Batchsize | 128 | 128 | 128 |
| Discount Factor | 0.99 | 0.99 | 0.99 |
| Replay Buffer Size | 100,000 | 100,000 | 100,000 |
| Gaussian Exploration Noise | 0.1 | 0.1 | 0.1 |
| Target Network Update Rate | 0.005 | 0.005 | 0.005 |

Table 4: Hyper-parameter of SAC under different environments.

|  | Point-Robot-Sparse | Reacher-Goal-Sparse | Swimmer-Sparse |
|---|---|---|---|
| Total Timesteps | 100,000 | 100,000 | 100,000 |
| Start Traning Timestep | 4,000 | 7,500 | 7,500 |
| Max Episode Length | 12 | 50 | 50 |
| Actor Network Hidden Layers | [256,256] | [256,256] | [256,256] |
| Critic Network Hidden Layers | [256,256] | [256,256] | [256,256] |
| Optimizer | Adam | Adam | Adam |
| Learning Rate | 0.0003 | 0.0003 | 0.0003 |
| Batchsize | 1024 | 1024 | 1024 |
| Discount Factor | 0.99 | 0.99 | 0.99 |
| Replay Buffer Size | 100,000 | 100,000 | 100,000 |
| Target Update Interval | 1 | 1 | 1 |

Table 5: Hyper-parameter of PPO under different environments.

|  | Point-Robot-Sparse | Reacher-Goal-Sparse | Swimmer-Sparse |
|---|---|---|---|
| Total Timesteps | 100,000 | 100,000 | 100,000 |
| Max Episode Length | 12 | 50 | 50 |
| Policy Network Hidden Layers | [64,64] | [64,64] | [64,64] |
| Value Network Hidden Layers | [64,64] | [64,64] | [64,64] |
| Optimizer | Adam | Adam | Adam |
| Learning Rate | 0.0003 | 0.0003 | 0.0003 |
| Batchsize | 64 | 64 | 64 |
| Discount Factor | 0.99 | 0.99 | 0.99 |
| GAE Parameter | 0.95 | 0.95 | 0.95 |
| Timesteps per Update | 2048 | 2048 | 2048 |
| Number of Epochs | 10 | 10 | 10 |
| Clipping Parameter | 0.2 | 0.2 | 0.2 |
| Value Loss Coefficient | 0.5 | 0.5 | 0.5 |

