# OpenReview forum: "CFlowNets: Continuous Control with Generative Flow Networks"
_ICLR.cc/2023/Conference — ICLR 2023 poster_

### Official Review · Reviewer_Rdez · 2022-10-21

**Confidence:** 5
**Correctness:** 3
**Technical Novelty And Significance:** 2
**Empirical Novelty And Significance:** 4
**Recommendation:** 8

**Clarity, Quality, Novelty And Reproducibility:**

The first paragraph of page 2 is not very clear. The authors should simply say that they use important sampling to approximate the integrals over in-flows and out-flows in the flow-matching GFlowNet constraint. The writing and clarifty could generally be improved.

There is interesting novelty in the paper, as explained above, somewhat in terms of the algorithm and significantly in terms of experiments (a first for continuous GFlowNets).


**Strength And Weaknesses:**

The main strength is that this is the first published paper exploring a continuous state-and-action with GFlowNets. A fixable weakness is that they claim that GFlowNets as previously published cannot handle continuous states and actions, which is not true (they were proposed in the Bengio et al 2021b paper, section 6). The authors also seem to have misunderstood the proposal in that paper. Nonetheless, Bengio et al 2021b was a purely theoretical paper and did not report experiments with continuous GFlowNets, so this is an important validation. In addition, the proposed approach to handle continuous variables is different from the one proposed by Bengio et al 2021b.

The mathematical derivations in page 3 and Theorem 1 are not really novel: they are identical or continuous versions of the equations in the GFlowNet papers, but replacing sums by integrals. This should be clarified to give a false impression of novelty. The real novelty is in how the authors propose to approximate the integrals and that is worth explaining in more detail. One concern I have is that importance sampling (used to approximate the integrals) is notoriously problematic in high dimension, due to rapidly increase variance of the estimator. Experiments where different numbers of dimensions are compared could be useful in this regard.

The authors present a series of experiments on control tasks in a low-dimensional state-space and a terminal reward, where the proposed formalism can be directly applied. This is a first and very interesting, along with the results showing greater diversity with GFlowNets (like earlier GFlowNet papers, but now in a control task with continuous variables), and even better rewards in two of the three tasks.

Detailed comments:

* In several places, e.g. par. 2 of sec. 1, 1st two lines of page 2, 1st par. of sec 2.2, the authors make false claims about GFlowNets for continuous states and actions. Bengio et al 2021b state very clearly that all the math shown in earlier sections can be applied to continous variables by replacing sums by integrals, and they propose using integrable densities and the detailed balance criterion to obtain a tractable training objective (with no need to approximate integrals by sums).

* The statement linking DAGs and MDPs (first sentence of 2.2) seems wrong: in general, the seequences of states associated with an MDP do not form a DAG (we can visit a state several times, i.e., form a cycle, in an MDP).

* The term "particle" is used on page 3 but not defined (I understand it comes from the GFlowNet paper but the readers of this paper may not have read them).

* The statement made on page 7 (end of 1st par. of sec. 5) about the proposal regarding continuous GFlowNets in Bengio et al 2021b is incorrect. That proposal does NOT require decomposing the continuous state into bins or discretizing it with continuous residuals. The math that is described can be applied with a purely continuous state. Hence there is no issue of exponential explosion as incorrectly claimed. I copy here some relevant sentences from that paper:
"However, for the most part one can replace these sums by integrals in case the states or actions are either continuous or hybrid (with some discrete components and some continuous components)."
"The challenge is to represent continuous densities on the output, with the need to both being able to compute the density of a particular value (say P (sxt+1 | sit+1, st)) and to be able to sample from it. Computing categorical probabilities and sampling from a conditional categorical is standard fare, so we only discuss the continuous conditional. One possibility is to parametrize sxt+1 | sit+1,st with a density for which the normalization constant is a known tractable integral, like the Gaussian."
"Other approaches include modeling the conditional density with an autogressive or normalizing flow model".


**Summary Of The Paper:**

A variants of GFlowNets with continuous state and action spaces is proposed, based the flow-matching conditions and importance sampling to approximate the sums over children and parents of each state. Low-dimenional control experiments are performed showing that a greater diversity of trajectories are obtained compared to a number of standard RL baselines, as would have been expected.

**Summary Of The Review:**

If the authors fix the false claims about previous work on GFlowNets with continuous valued states and actions (which is localized in a few sentences and should be easy to do without changing the main messages of the paper), the paper should be published.

** update: ** given the changes promised by the authors, I have increased my recommendation from 6 to 8.

The experiments on control tasks are interesting, and a first for continuous GFlowNets, so quite interesting as experimental proof that GFlowNets can be applied in continuous domains.

---

> ### Author Response · Authors · 2022-11-19
> **Response to Reviewer Rdez (Part 1)**
>
> Dear Reviewer Rdez,
>
> Thank you for your comments. We will make the modifications to the paper for a better understanding based on your suggestions.
>
> [**Comment 1:**] *The mathematical derivations in page 3 and Theorem 1 are not really novel.*
>
> [**Response for Comment 1:**] Thanks for your good suggestion, we modified the word ''propose'' to ''extend'' in the main contributions to avoid misunderstandings, as follows:
>
> *''We extend the theoretical formulation and flow matching theorem of previous GFlowNets to continuous scenarios.''*
>
> [**Comment 2:**]  *One concern I have is that importance sampling (used to approximate the integrals) is notoriously problematic in high dimension, due to rapidly increase variance of the estimator. Experiments where different numbers of dimensions are compared could be useful in this regard.*
>
> [**Response for Comment 2:**] We add a new autonomous driving scene with higher dimensions to verify the effectiveness of our algorithm. Indeed, when the dimension increases, the error of the flow matching approximation will also increase. However, the learning efficiency of reinforcement learning algorithms will also decrease in high dimensions. Therefore, our algorithm still has an advantage in comparison, especially in exploring this indicator.
>
> [**Comment 3:**] *In several places, e.g. par. 2 of sec. 1, 1st two lines of page 2, 1st par. of sec 2.2, the authors make false claims about GFlowNets for continuous states and actions. Bengio et al 2021b state very clearly that all the math shown in earlier sections can be applied to continous variables by replacing sums by integrals, and they propose using integrable densities and the detailed balance criterion to obtain a tractable training objective (with no need to approximate integrals by sums).*
>
> [**Response for Comment 3:**] We really appreciate your good advice. We rewrite par.2 of sec.1 as the following:
>
> *''GFlowNets structure the state transitions of trajectories into a directed acyclic graph (DAG) structure. Each node in the graph structure corresponds to a different state, and actions correspond to transitions between different states, that is, an edge connecting different nodes in the graph. For discrete tasks, the number of nodes in this graph structure is limited, and each edge can only correspond to one discrete action. However, in real environments, the state and action spaces are continuous for many tasks, such as quadrupedal locomotion (Kohl Stone, 2004), autonomous driving (Kiran et al., 2021; Shalev-Shwartz et al., 2016; Pan et al., 2017), or dexterous in-hand manipulation (Andrychowicz et al., 2020). Moreover, the reward distributions corresponding to these environments may be multimodal, requiring more diversity exploration. The needs of these environments closely match the strengths of GFlowNets. (Bengio et al., 2021b) proposes an idea for adapting GFlowNets to continuous tasks by replacing sums with integrals for continuous variables, and they suggest the use of integrable densities and detailed balance (DB) or trajectory balance (TB) Malkin et al. (2022) criterion to obtain tractable training objectives, which can avoid some integration operations. However, this idea has not been verified experimentally.''*

---

> > ### Author Response · Authors · 2022-11-19
> > **Response to Reviewer Rdez (Part 2)**
> >
> > [**Comment 4:**] *The statement linking DAGs and MDPs (first sentence of 2.2) seems wrong: in general, the sequences of states associated with an MDP do not form a DAG (we can visit a state several times, i.e., form a cycle, in an MDP).*
> >
> > [**Response for Comment 4:**]  Incorrect statement about linking DAGs and MDPs have been removed.
> >
> > [**Comment 5:**]  *The term "particle" is used on page 3 but not defined (I understand it comes from the GFlowNet paper but the readers of this paper may not have read them).*
> >
> > [**Response for Comment 5:**] Thanks for your suggestion. We added a reference to the particle. We did not describe in detail due to space constraints.  The revised part on page 3 is as the following:
> >
> > *"For each trajectory $\tau$, the associated flow $F(\tau)$ contains the number of particles (Bengio et al., 2021b) sharing the same path $\tau$."*
> >
> > [**Comment 6:**]  *The statement made on page 7 (end of 1st par. of sec. 5) about the proposal regarding continuous GFlowNets in Bengio et al 2021b is incorrect.*
> >
> > [**Response for Comment 6:**]  Thanks for your comments. Incorrect statements about the proposal regarding continuous GFlowNets in (Bengio et al., 2021b) are updated.  The revised part of sec. 5 on page 7 is as the following:
> >
> > *"In (Bengio et al., 2021b), an idea is proposed for adapting GFlowNets to continuous tasks by replacing sums with integrals for continuous variables. (Malkin et al., 2022) and (Bengio et al., 2021b) propose detailed balance (DB) and trajectory balance (TB) objectives, which use parametric forward and backward policies in the objective function. These new objective functions do not require evaluating the flow model on multiple parents of a state, which is more efficient, especially for high-dimensional environments. (malkin et al., 2022) and (Bengio et al., 2021b) mentioned that these objective functions could also be used in continuous scenarios by replacing the policy likelihoods in the objective with probability densities. A possible disadvantage is that it is not easy to estimate $P_F$ and $P_B$ in a continuous environment, since the state space is much larger than in a discrete scenario, and a small error in modeling probability densities can greatly affect the final performance. How to combine DB and TB with CFlowNets will be a worthy future work."*

---

> ### Author Response · Authors · 2022-11-21
> **Response to Reviewer Rdez**
>
> We are glad that the reviewer appreciates our attempt, and sincerely thank the reviewer for raising the score. Your constructive comments help improve our paper.

---

### Official Review · Reviewer_T9R9 · 2022-10-21

**Confidence:** 5
**Correctness:** 3
**Technical Novelty And Significance:** 3
**Empirical Novelty And Significance:** 3
**Recommendation:** 8

**Clarity, Quality, Novelty And Reproducibility:**

The paper is fairly clear, and this is as far as I know the first attempt to make GFlowNets continuous. I am confident that I could reproduce these results.

**Strength And Weaknesses:**

### Strengths
- The paper proposes a sensible, if expensive, approach to making GFlowNets compatible with continuous state spaces
- The experimental results are encouraging, and compatible with previous work suggesting that GFNs are capable of covering states spaces much better than RL
- The paper is generally well written, and although some sections could be improved, the overall message is clear

### Weaknesses
- GFlowNet is a new framework, and isn't deeply understood, mathematically or empirically. There isn't much, if anything, in the paper which suggests that the agents are doing better _because of GFN_.
  - It's not demonstrated that the learned flow distributions $F_\theta$ really model the theoretically proposed flows
  - It's not demonstrated that the current formulation converges
  - The proposed bound is an interesting result but little is known of Lipschitz constants of flow functions (and that's without going into how large DNN Lipschitz constants can get), which makes it hard to relate the bound to empirical quantities.
- Some of the evaluations done in the paper seem partially incorrect (see below)


### Specifics

Eq (11) is a bit weird, the theorem prompt asks us to consider $\hat F (s,a)$, yet the equations refer to $\hat F(s \to s')$ and $\hat F(s)$. It would be good to clarify exactly what is what.

The indicator function in Eq (13) seems unnecessary. $R(s_t)$ is by definition 0 for all $s_t \neq s_f$, and ${\cal C}(s_f)$ is the empty set.

> For continuous tasks, it impossible to access all state-action pairs to calculate the continuous inflows and outflows.

It is impossible to enumerate all the children of a state, but it is possible in some cases to take integrals over well defined things like Gaussian distributions. It would be nice if this were addressed in the paper.

> we sample an action probability buffer based on the forward-propagation of CFlowNets, from which we can sample an action with probability proportional to the reward.

This isn't quite the correct wording, and this formulation is repeated multiple times in the paper. GFlowNets (and actually CFlowNets as well, presumably) sample _terminal states_ with probability proportional to their reward. They do not sample _actions_ with probability proportional to any reward (unless in the bandit setting).

Sec 4.3 is a bit confusing.

>  we should find the parent state first.

parent _states_? There should be many.

> a transaction deep neural network

I'm not sure what a _transaction_ DNN is. Do you mean a transition DNN? Transient?

> [$G$ has] $(s_{t+1}, a_t)$ as the input while $s_t$ as the output, and train this network based on $B$ with the MSE loss.

This assumes that taking action $a_t$ in $s_t$ is the _only_ way to get to $s_{t+1}$. This is not a totally unreasonable assumption, but it seems relatively easy to break: imagine a single joint at $\theta\in[0,180]$ degrees which moves an arm by $a\in[-180,180]$ degrees, now imagine that there is a wall such that moving the joint past $\theta>110$ degrees makes the arm block against the wall. Then for the state $s_{t+1}=\theta=110$, there are many possible parents with the same action, e.g. $(\theta=90, a=20)$ and $(\theta=100, a=20)$ are both valid parents with the same $a_t$.

Either way this assumption should be clearly stated.


> All of these improved policy gradient methods can be classified as aiming at maximizing reward

Yes, although to be fair most modern PG implementations include some form of entropy regularization (inducing entropy on the trajectory distribution). This is in some sense the basis of control-as-inference/MaxEnt methods, by maximizing the entropy some cover of the state space is induced.

> what SAC learns is not the true distribution of strictly proportional return, [..] This is different from being directly proportional to reward.

The wording here isn't quite correct. In the gaussian approximation, no, SAC doesn't exactly learn to be proportional to return, but in the general case, yes. But that is not what differentiates SAC and GFN: SAC (or control as inference) learns something like $p(\tau)\propto G(\tau)$ while GFN learns $p(x) \propto G(\tau)$ when $\tau=(s_1,...,s_T)$ ends at $x$, or to relate the two, learns $\sum_{\tau:s_T=x} p(\tau) = G(\tau_{s_T=x})$. GFN considers all possible trajectories that lead to a terminal state, while SAC (and PG algorithms in general) are trajectory and return-centric: they do not "care" about the specific terminal state which the agent reaches.


> Figure 2: Reward distributions on Point-Robot-Sparse Task

I'm not sure I understand Figure 2. If I understand correctly, in Point-Robot, an agent is contained within a 2D surface and must navigate using continuous actions to reach some goal, here (10,5) or (5,10). "Each time the agent can choose to take a step from any angle to the upper right." This suggests to me that the action is the angle, or more accurately, the agent moves in the direction of the vector (1,1) rotated by $a \pi/4$ radians. After checking the supplementary material this seems roughly correct, the agent moves by $(\cos(\theta), \sin(\theta))$ where $\theta = (a + 1)\pi / 4$.

What Figure 2 seems to show is $F(s,a)$, or $V(s')$ for the RL methods, and $R(s')$ for the "True Reward" _if the agent were to terminate_ (since we are in the sparse setting, I'm assuming this means the agents only get terminal rewards). This seems incorrect, and I'm not sure why the y axis is called "Estimated Reward Value".

I see two problems here:
- I see no reason for flow matching (even continuous) to predict $F(s,a) = R(s')$  when $s'$ is not terminal, this is because with FM there is no notion of preference over paths (see Figure 10 of Bengio et al. 2021a). If this plot was obtained by setting `cnt_step` to 11, this should be explicit.
   - Similarly for the RL baselines, I see no reason for RL agents to prefer going up or right at (7,7) proportionally to $R(s')$, in fact, the solution found by TD3 might be just fine, it wants to go up to get closer to (5, 10).
   - Again for the RL baselines, it's not clear why $V(s')$ should be equal to $R(s')$ unless we're terminating.
   - The straight line for DDPG is honestly suspect. Looks like a bug?
- As far as I can tell in the current code the agent never sees the current timestep. This is a problem since, as I point out, the plot of Figure 2 only makes sense if (7,7,`cnt_step=11`) is the current state, and more generally, any state has multiple paths which lead to it. Depending on the current timestep the flow predictions for a state _must_ be different (think about the (7,7) example, if there is only 1 action left, then the flow F(s,a) is equal to the reward of the next state, but if there are 2 actions left, then there are many more accessible states and rewards, and so F(s,a) will be larger). As suggested by Bengio et al 2021b S3.3.1, augmenting the state with the current timestep automatically induces a DAG and would make much more sense here.

If this was truly a terminal reward plot, then it would be a bit reassuring, but it's not obvious to me that this implies that the flow function is fit well in the rest of the state space.



Some typos: theoritical -> theoretical, "a cyclic will occur" -> a cycle will occur, "need to sum" -> needs to sum, "the set contains all" -> the set that contains all, "that starting in s0 and ending in s" -> that starts in s0 and ends in s, "to form a cyclic" -> to form a cycle, "we can modified equation 19" -> we can modify equation 19, "has good sample-efficient property" -> has good sample efficiency.

**Summary Of The Paper:**

This paper proposes a continuous formulation of GFlowNet, for both the action and state spaces, by converting those spaces into continuous spaces and converting the sums of flow-matching into integrals. This yields a continuous flow matching objective which in practice is estimated by Monte Carlo integration.
The authors show an error bound on the accuracy of the MC flow estimation which decreases loglinearly in the number of samples, provided the real flow function is Lipschitz continuous.
This method is then benchmarked against three standard continuous control tasks, where it beats standard RL methods on 2 of them, and in all 3 cases appears to cover much more of the state space than any baseline.


**Summary Of The Review:**

I am mitigated, on one hand this is a cool and necessary extension of the GFlowNet framework towards continuous domains, on the other hand there is little evidence that the method does what is suggested. Perhaps it does fit flows and learn the right thing, but since relevant quantities are not measured empirically it is impossible to know.

~As is I think the paper needs improvements in order to be accepted, improvements that I think could be addressed in a rebuttal time-length.~

Update: The updated revision of the paper is an improvement and makes many things clearer, and so I am changing my score from 5 to 8.

---

> ### Author Response · Authors · 2022-11-19
> **Response to Reviewer T9R9 (Part 1)**
>
> Dear Reviewer T9R9,
>
> Thank you for your comments. We will make the modifications to the paper for a better understanding based on your suggestions.
>
> [**Comment 1:**] *It's not demonstrated that the learned flow distributions really model the theoretically proposed flows.*
>
> [**Response to Comment 1:**]  Thanks for your comments. We add more results on the learned reward distribution in the appendix to show that our algorithm can learn the desired flow distributions well.
>
> [**Comment 2:**] *Eq (11) is a bit weird, the theorem prompt asks us to consider, yet the equations refer to and. It would be good to clarify exactly what is what.*
>
> [**Response to Comment 2:**] Thanks for your suggestion. Eq (11) is revised for better understanding in the revised version.
>
> [**Comment 3:**] *For continuous tasks, it is impossible to access all state-action pairs to calculate the continuous inflows and outflows.*
>
> [**Response to Comment 3:**] Thanks for your comments. We modified this description into:
>
> *"For continuous tasks, it is usually difficult to access all state-action pairs to calculate continuous inflows and outflows."*
>
> [**Comment 4:**] *we sample an action probability buffer based on the forward-propagation of CFlowNets, from which we can sample an action with probability proportional to the reward.*
>
> [**Response to Comment 4:**] All related descriptions are revised.
>
>
> [**Comment 5:**]  *"we should find the parent state first." parent states? There should be many.*
>
> [**Response to Comment 5:**]  we modified state --> states.
>
> [**Comment 6:**] *I'm not sure what a transaction DNN is. Do you mean a transition DNN? Transient?*
>
> [**Response to Comment 6:**] Thanks for your comments. This is really confusing. We modified the transaction deep neural network into the retrieval neural network for better understanding. Since this neural network is used for state retrieval.
>
> [**Comment 7:**] *The assumption that taking action $a_t$  in $s_t$ is the only way to get to $s_{t+1}$ should be clearly stated.*
>
> [**Response to Comment 7:**] Thanks for your suggestions. The related assumption is added. The corresponding necessity and rationality are discussed in Appendix A.

---

> > ### Author Response · Authors · 2022-11-19
> > **Response to Reviewer T9R9 (Part 2)**
> >
> > [**Comment 8:**] *All of these improved policy gradient methods can be classified as aiming at maximizing reward*
> >
> > [**Response to Comment 8:**] Thanks for your comments. We revised this description as:
> >
> > *"Most of these improved policy gradient methods can be classified as aiming at maximizing reward"*
> >
> > [**Comment 9:**] *"what SAC learns is not the true distribution of strictly proportional return, [..] This is different from being directly proportional to reward." The wording here isn't quite correct.*
> >
> > [**Response to Comment 9:**]  The related descriptions have been modified accordingly, which is revised as follows:
> >
> > *"1) SAC selects actions by a Gaussian policy, which is less expressive than using a general unnormalized action p.d.f. $F(s,a)$;
> > 2) In the general case, SAC learns to be proportional to the long-term return, which generates the trajectory distribution satisfying $p(\tau) \propto R(\tau)$ with $R(\tau)$ is the return of $\tau$. CFlowNets considers all possible trajectories that lead to a terminal state $s_f$, and learn the policy to generate $s_f$ with $p(s_f) \propto R(s_f)$."*
> >
> > [**Comment 10:**] *Reward distributions on Point-Robot-Sparse Task.*
> >
> > [**Response to Comment 10:**]  It is worth noting that for our continuous tasks, the terminating state space is a range. That is, the range can be transferred to the final goal in one action. Hence, we pick some state points around this terminating state range to calculate the learned flow distribution. For example, our goal is (5, 10) and (10, 5), hence we plot the reward distributions at (7, 7), (8, 4), (4, 8).
> >
> > These distributional results are plotted by varying the actions while fixing the state of at these selected points based on the learned flow network. These distributions are plotted during inference process, it is difficult to plot the learned distribution during training since they go to different positions at cnt-step=11 for our continuous tasks. But the points we choose is the relatively dense area at cnt-step=11. To some extent, it can also be understood as the distribution result at cnt-step=11.
> >
> > [**Comment 11:**] *As for "The straight line for DDPG is honestly suspect. Looks like a bug?"*
> >
> > [**Response to Comment 11:**]
> > We double-checked the code, and there is no bug but we found DDPG in this task is prone to overfitting with 1e6 total timesteps. We hence added reward distribution results in Figure 11 with 2e4 total timesteps to avoid overfitting. We can see that DDPG works in Figure 11.
> >
> > [**Comment 12:**] *As for "(think about the (7,7) example, if there is only 1 action left, then the flow F(s,a) is equal to the reward of the next state, but if there are 2 actions left, then there are many more accessible states and rewards, and so F(s,a) will be larger)"*
> >
> > [**Response to Comment 12:**]  Yes, all the learned distributions are after normalization processing. In many case, $F(s,a)$ is larger than the reward, but they have the same trend.

---

> ### Author Response · Authors · 2022-11-26
> **Response to Reviewer T9R9**
>
> Dear reviewer T9R9,
>
> We greatly appreciate your valuable suggestions, which have helped us improve the quality of the paper significantly. We have added many visualizations in the revised version to demonstrate the learned flow distributions model of the theoretically proposed flows. More experiment results and higher dimensional scenarios are added to show that the algorithm can converge well. In addition, we have made many revisions based on your constructive suggestions to make this paper more rigorous. Our work is the first experimental validation of continuous GFlowNets.  Please let us know if you have other questions or comments.
>
> Since the discussion window between reviewers and authors is ending, we sincerely look forward to your reevaluation of our work and would very appreciate it if you could raise your score to boost our chance of more exposure to the community. Thanks a lot!

---

> > ### Comment · Reviewer_T9R9 · 2022-12-05
> > **Good update**
> >
> > Pardon my lack of engagement, life kept me busy. I have looked at your response and the revision of the paper, and they clarify many things. I will raise my score.

---

> > > ### Author Response · Authors · 2022-12-06
> > > **To Reviewer T9R9**
> > >
> > > Thank you so much for taking the time to further evaluate the value of our work! Your constructive comments have greatly helped our paper to be improved!

---

### Official Review · Reviewer_Y6ub · 2022-10-23

**Confidence:** 4
**Correctness:** 2
**Technical Novelty And Significance:** 3
**Empirical Novelty And Significance:** 4
**Recommendation:** 6

**Clarity, Quality, Novelty And Reproducibility:**

The paper is well written, and very clear. The authors provided the source code (in the supplementary material), along with all the hyperparameters used in the Appendix.

Some extensions of GFlowNets to hybrid state and action spaces have been introduced in (Bengio et al., 2021b), including a method to handle continuous states and actions via a (normalized) distribution over actions. Here, CFlowNets are novel in the sense that they do not require an explicit normalized distribution as a policy, but is defined similarly to GFlowNets in terms of continuous flows. Moreover, to the best of my knowledge, this work is the first to empirically show the effectiveness of flow networks on continuous control tasks.

---

*Yoshua Bengio, Tristan Deleu, Edward J. Hu, Salem Lahlou, Mo Tiwari, and Emmanuel Bengio. Gflownet foundations, 2021b.*

**Details Of Ethics Concerns:**

No Ethics Concerns

**Strength And Weaknesses:**

**Strengths**: The generalization to continuous state and action spaces proposed here is a very important step towards making GFlowNets applicable to a broader setting beyond discrete states. These generalizations are for the most part natural extensions of prior results, replacing summation by integration anywhere applicable, which makes this new formulation very appealing. And while some extensions to hybrid states and actions have been proposed in the past (Bengio et al., 2021b), CFlowNet is the first work to show empirical evidence of the effectiveness of flow networks in continuous settings. The practical implementation of CFlowNets via sample actions and the transaction network is also well thought, and the theoretical guarantees about the approximation error (Theorem 2) is appreciated to ensure that these approximations are valid.

**Weaknesses**: Unfortunately, this paper falls short in big ways theoretically, with many approximations in their formulation of continuous flows. The authors are too often ignoring some technical aspects, as well as making implicit assumptions that are crucially missing.

 1. The very first definition (Definition 1) of the continuous state flow as $F(s) = \int_{\tau:s\in\tau} F(\tau)d\tau$ is ill defined: what guarantees do we have that this integral is properly defined and finite? I guess you would have to assume that $F$ defines a finite measure over the space of complete trajectories $\tau$. This is only one example, but none of the integrals used throughout the paper have any guarantee to exist.
 2. Related to 1., the forward transition probability and backward transition probability in equations (5) & (6) are only properly defined if all the terms involved are well defined and finite.
 3. The formulation of CFlowNet requires a generalization of the notion of parents and children to trajectories over continuous states (as opposed to parents and children in a DAG for (discrete) GFlowNets). However the definition of parents of a state, preserving the acyclicity constraint, is incorrect (and similarly for children). In the paper (p.3), the parents $\mathcal{P}(s_{t})$ of $s_{t}$ is defined as the set of states $s$ such that (1) $T(s, a\in A) = s_{t}$ an action $a$ could make a transition from $s$ to $s_{t}$, and (2) $T(s, a\in A) \notin \mathrm{set}(s_{0}, \ldots, s_{t-1})$ the state cannot be transferred to a preexisting state, otherwise a cyclic would occur. This implicitly assumes that we are taking a specific (partial) trajectory from $s_{0}$ to $s_{t}$, and we don't want to "wrap around" that particular trajectory with any action. However, this condition should be satisfied **for any** trajectory from $s_{0}$ to $s_{t}$, not just a single trajectory. The definition of children is also incorrect and should be defined **for any** trajectory from $s_{0}$ to $s_{t}$, and this has very serious implications later in the paper. I am including an **Example** below to explain why this criterion is incorrect.
 4. A big consequence of 3. is that Remark 2 in the paper is incorrect, and you must handle the acyclicity carefully, even in practice (this is not simply for "theoretical convenience" as claimed in this Remark. See the **Example** below (point 2). A similar claim is made in Section 4.4 too (the measure of $\mathcal{A}$ is constant).
 5. In Assumption 1, you assume that $a \mapsto F(s, a)$ is a Lipschitz continuous, with Lipschitz constant $L_{s}$. However since the state space is itself continuous, how can we guarantee that $\sup_{s\in\mathcal{S}}L_{s}$ is finite?
 6. You implicitly make the assumption that the state $\mathcal{A}$ has finite measure (using $\mu(\mathcal{A})$), and has finite support (using $\mathcal{diam}(\mathcal{A})$). Moreover, the action space $\mathcal{A}$ must be a measured space defined ahead of time, so that $\mu(\mathcal{A})$ is a fixed constant, and therefore it is incorrect to claim that $K/\mu(\mathcal{A})$ may be considered as a hyperparameter $\lambda$ (Section 4.4). Otherwise, if $\lambda$ is truly a free hyperparameter, what prevents you to set it to $1$?
 7. According to (Bengio et al., 2021b), a GFlowNet defines a distribution proportional to the rewards **over terminating states** (to borrow their naming conventions, i.e. the parent states of the terminal states $s_{f}$). This is not what is claimed in this paper. For example Section 1: "*In contrast, the training goal of GFlowNets is to approximately sample candidate actions with probability proportional to a given reward function*", Section 4.1: "*we can sample an action with probability proportional to the reward*", Section 5: "*generating policies that sample objects through discrete action sequences with probabilities proportional to a predefined reward function*", Section 5: "*for CFlowNets [...] the probability of an action being sampled is proportional to the corresponding reward*". There seems to be some confusion between the distribution proportional to the rewards found by GFlowNets/CFlowNets and the policy (which, according to equation (5), is proportional to the continuous flow). While the flow itself depends on the reward, it is misleading to claim that the policy samples actions proportionally to the rewards.

---

**Example**: Consider the 2D point-mass environment, where the agent moves on a square of size 2 around it with a certain angle. The state space is $\mathcal{S} = \mathbb{R}^{2}$ is any point on the plane, and the action space $\mathcal{A} = [0, 2\pi]$ (the direction in which the agent moves). Given a state $s = [x, y]$, and an action $a = \theta$, the environment transitions to a new state $T(s, a) = s' = [x', y']$, where $x' = x + \mathrm{clip}(\sqrt{2}\cos \theta, -1, +1)$ and $y' = y + \mathrm{clip}(\sqrt{2}\sin \theta, -1, +1)$. We assume that we start at the initial state $s_{0} = [0, 0]$.

 - Since the actions are completely reversible (we can apply action $\theta$ and then $\theta + \pi \mod 2\pi$ to get back to the same state), the set of parents $\mathcal{P}(s_{t}) = \emptyset$ for any state $s_{t}$ is empty (except all the states on the square of size 2 around $s_{0}$, whose only parent would be $s_{0}$), because if $s\in\mathcal{P}(s_{t})$ such that $T(s, a) = s_{t}$ and there exists $s'$ such that $T(s', \theta) = s$ ($s' \rightarrow s$ along the trajectory), then we have $T(s, \theta + \pi \mod 2\pi) = s'$, which violates the condition of $s\in\mathcal{P}(s_{t})$. This is a very restrictive notion on parents; but let's imagine for the remainder of this example that we allow situations where the set of such states is of measure $0$ instead.

 - Consider the state $s_{t} = [2, 0]$. We have many ways to arrive in $s_{t}$ with a trajectory of length 2; for example $[0, 0] \rightarrow [1, 0] \rightarrow [2, 0]$, or $[0, 0] \rightarrow [1, 1] \rightarrow [2, 0]$, etc... Effectively, this means that the set of parents $\mathcal{P}(s_{t}) \subset \mathrm{set}([1, y] : y \in [-1, 1])$. In particular, since for obvious acyclicity reasons the set of children of $s_{t}$ must be distinct from the set of its parents, the set of valid actions satisfies $[3\pi/4, 5\pi/4] \not\subset \mathrm{set}(a\in\mathcal{A} : T(s_{t}, a) \in \mathcal{C}(s_{t}))$. In other words, the whole interval $[3\pi/4, 5\pi/4]$ are invalid actions from $s_{t}$, and this does not have measure $0$ in $\mathcal{A}$, and therefore we can't ignore invalid actions contrary to what is claimed in Remark 2. The situation here is even worse, because we only considered trajectories of length 2 so far; in fact, we can find trajectories so that the parents of $s_{t}$ are all the states on the square of size 2 around $s_{t}$, meaning that $\mathcal{C}(s_{t}) = \emptyset$.

 - Consider the state $s_{3} = [3, 0]$. If we follow the definition of $P(s_{3})$ in the paper, we need to find states $s_{2}$ such that $T(s_{2}, a\in \mathcal{A}) = s_{3}$ and $T(s_{2}, a\in\mathcal{A}) = \mathrm{set}(s_{0}, s_{1}, s_{2})$. For similar reasons as above, a parent state will have the form $s_{2} = [2, y]$, with $y \in [-1, 1]$. If we only consider a specific trajectory $(s_{0}, s_{1}, s_{2})$ (as is again implied by the argument on the set of measure 0 in Remark 2), then we can take the trajectory $[0, 0] \rightarrow [1, y] \rightarrow [2, y]$; modulo the first remark above about sets of measure $0$, state $s_{2}$ satisfies the criterion for being a parent of $s_{3}$. However, we have other trajectories $[0, 0] \rightarrow [1, y'] \rightarrow [2, y]$, for any $y' \in [0, y]$ that are also trajectories, and are distinct from the single trajectory we had to consider before to satisfy the criterion for $\mathcal{P}(s_{3})$, and in those case, we can reach $[1, y']$ from $[2, y]$. This would violate the acyclicity condition, because we can find for example a cycle of the form $[1, y] \rightarrow [2, y] \rightarrow [1, 0] \rightarrow [2, 0] \rightarrow [1, y]$. Moreover, the set of such cycles is not of measure 0. That's why the condition for parents should be at least over **all possible trajectories** leading to $s_{3}$.

Note that this example probably suggests that some conditions on the MDP are necessary for the framework of CFlowNets to be valid (similar to how the DAG assumption was necessary in GFlowNets).

---

*Yoshua Bengio, Tristan Deleu, Edward J. Hu, Salem Lahlou, Mo Tiwari, and Emmanuel Bengio. Gflownet foundations, 2021b.*

**Summary Of The Paper:**

This paper extends the work of (Bengio et al., 2021a) on Generative Flow Networks (GFlowNets), which was limited to discrete state and actions spaces, to continuous control tasks. The authors introduce the notion of continuous flow $F$, and they formulate the conditions required by this function $F$ to correspond to a properly defined Markovian flow, via a generalization of the flow-matching condition of (Bengio et al., 2021b) to continuous states and actions. This new framework is called CFlowNets. Similar to GFlowNets, CFlowNets  While this provides theoretical guarantees, this condition involves integrating over the whole action space, which is typically impractical; fortunately, the authors devise a practical approximation of the flow-matching condition that only involves sample actions only to estimate the outflow, as well as a separate neural network, called the transaction network, that is responsible for estimating the inflow. The authors also provide theoretical guarantees on the soundness of their approximation. Finally, CFlowNets have been applied successfully to 3 different continuous control problems, both in terms of estimating the reward distribution accurately, and in terms of raw performance and exploration on sparse reward environments.

---

*Emmanuel Bengio, Moksh Jain, Maksym Korablyov, Doina Precup, and Yoshua Bengio. Flow network based generative models for non-iterative diverse candidate generation, 2021a.*

*Yoshua Bengio, Tristan Deleu, Edward J. Hu, Salem Lahlou, Mo Tiwari, and Emmanuel Bengio. Gflownet foundations, 2021b.*

**Summary Of The Review:**

I am very hopeful with this submission, and I believe this could be a very important step toward applying flow networks to continuous control problems. The fact that the authors showed empirically that CFlowNets offer an advantage on exploratory tasks is also an important evidence. I have unfortunately not been able to thoroughly review the proofs in the Appendix, as well as the source code provided, due to the short period of time for the reviews.

Despite my enthusiasm, I strongly believe that in its current state the paper cannot be accepted. The paper is making too many approximations on the theory, and those missing details are crucial. The list I provided is certainly not exhaustive, and most of the theory part of this submission must be reworked with care. That's why I can't say that the paper "has minor issues that only require small changes" (in **Correctness**), because this requires major changes, albeit technical.

Because of this, I am currently recommending borderline reject.

---

> ### Author Response · Authors · 2022-11-19
> **Reponse to Reviewer Y6ub**
>
> Dear Reviewer,
>
> Thank you very much for your many good suggestions, which are of great help to us in improving the quality of the paper.
>
> [**Comment 1:**] *Whether the integral in Definition 1 exists.*
>
> [**Comment 2:**] *Whether equations 5 and 6 hold.*
>
> [**Comment 5:**] *Whether the Lipschitz constant finite*
>
> [**Response to Comments 1,2,5**] We put the assumption that the flow network is Lipschitz continuous at the beginning, because it is assumed that the network is Lipschitz continuous, as long as the Lipschitz constant is bounded, then the flow network can be integrated. We have shown that the Lipschitz constant is bounded by experiments, please refer to Appendix A. In addition, Lipschitz continuous is a common assumption of neural networks, just some quick examples: [1]-[4] all use this assumption to prove the convergence of algorithms. We discuss the necessity and rationality of the continuous assumption in Appendix A.
> [1] Du, Simon, et.al., . “Gradient descent finds global minima of deep neural networks.” In International conference on machine learning, pp. 1675-1685. PMLR, 2019.
> [2] Jacot, Arthur, et.al., “Neural tangent kernel: Convergence and generalization in neural networks.” Advances in neural information processing systems 31 (2018).
> [3] Allen-Zhu, et.al., ”A convergence theory for deep learning via over-parameterization.” In International Conference on Machine Learning, pp. 242-252. PMLR, 2019.
> [4] Alistarh, Dan, et.al., ”The convergence of sparsified gradient methods.” Advances in Neural
> Information Processing Systems 31 (2018).
>
> [**Comment 3:**] *Unreasonable constraints in parent and child sets.*
>
> [**Response to Comment 3:**] We greatly appreciate your constructive comments. We removed the constraints in the definitions of parent and children set and added an assumption to guarantee acyclicity. The corresponding necessity and rationality are discussed in Appendix A. We have carefully revised the full text and related proofs.
>
>
> [**Comment 4:**] *Remark 2 is wrong.*
>
> [**Response to Comment 4:**]  Remark 2 has been deleted. Since we added assumptions, this remark is no longer needed.
>
> [**Comment 6:**] *About hyperparameter lambda*
>
> [**Response to Comment 6:**]  Many thanks for your question.  In fact, in our experiment, the value of lambda is set to 1. What we wanted to express before is that because we have performed a shaping operation on the reward, the error caused by the lambda can be replaced by a suitable reward shaping operation. So lambda is more like a shaping operation. Our previous statement is really hard to understand. We revise the description as the following:
>
> *''Note that in many tasks we cannot obtain exact $\mu(\mathcal{A})$. For such tasks, we can directly set $\lambda$ to 1, and then adjust the reward shaping to ensure the convergence of the algorithm. A commonly used reward shaping method is to multiply the reward by a constant and adjust the reward to an appropriate range to ensure better convergence. Therefore, after setting $\lambda$ to 1, a reasonable reward shaping operation can also compensate for the influence of $\lambda$ error.''*
>
> [**Comment 7:**] *about wrong statements*
>
> [**Response to Comment 7:**]  - Thank you very much for pointing out these errors, we have revised the relevant descriptions as follows:
>
> *"In contrast, the training goal of GFlowNets is to approximately sample candidate actions with probability proportional to a given reward function"* -->
> *"In contrast, the training goal of GFlowNets is to define a distribution proportional to the rewards over terminating states, i.e., the parent states of the final states."*
>
> *"we can sample an action with probability proportional to the reward"* --> deleted
>
> *"generating policies that sample objects through discrete action sequences with probabilities proportional to a predefined reward function"* --> *"by generating a distribution proportional to the rewards over terminating states"*
>
> *"for CFlowNets [...] the probability of an action being sampled is proportional to the corresponding reward"* --> *"the training goal of CFlowNets is to define a distribution proportional to the rewards over terminating states, resulting in more diverse trajectories that are beneficial for exploring the environment."*
>
> [**Comment Example:**] *about 2D point-mass environment*
>
> [**Response to Comment Example:**] Thank you very much for the very good example. Inspired by the example you gave, we give the example in Figures 4 and 5 to help readers understand.

---

> > ### Comment · Reviewer_Y6ub · 2022-12-01
> > **Post-rebuttal response**
> >
> > Thank you for your response. It addresses many of my concerns, and I am happy to see the paper has been greatly improved.
> >
> > Regarding Assumption 2 (Lipschitz continuity), its purpose was mainly to prove Theorem 2, and I think it made sense. My concern was more related to the guarantees that the Lipschitz constant would not blow up as it is a function of the state, and cause issues in Theorem 2. However, in the revision, Assumption 2 is now claimed to be used also for guaranteeing the existence (and being finite) of the different integrals throughout the paper (as explained in Appendix A.2). However, a function being Lipschitz does not guarantee you that the integral would be finite (for example, $F(s, a) = 1$ on an unbounded state and action space). You need these integrals to be finite, because you are using them to further define $P_{F}(s_{t+1}\mid s_{t})$ in Equation 8 for example. Therefore, Assumption 2 alone is not sufficient to address my concern related to the existence of finite integrals (points 1 & 2 in my review). The empirical verification of the Assumption (Figure 6) is greatly appreciated though.
> >
> > Regarding the rationality of Assumption 1 (acyclicity), I think the claim "*This assumption is reasonable because for many continuous environments it is difficult to form cycles in trajectories without special constraints.* in Appendix A.1 (and also in the discussion with Reviewer y8C2, regarding almost sure acyclicity) is a bit too strong. I don't think it is that hard to find standard RL environments where we can have cycles with non-zero measure. Pendulum (without wall) has the property that you can only go from one state to another through a single trajectory, so it is a very specific environment. Even the example of Pendulum with walls is not enough to highlight the possibility to have cycles, because there is again a single way to go from any state to any other (if we allow "almost-sure" acyclicity as argued in the discussion with Reviewer y8C2). A more convincing example to me to the possibility to have cycles would be Acrobot, because there are many different ways to reach the same state from any starting position (so the set of cycles would not have measure zero). But Acrobot is one example, it would work for CartPole, MountainCar too.
> >
> > I will be happy to increase my score once these last points are discussed.

---

> > > ### Author Response · Authors · 2022-12-02
> > > **Response to Post-rebuttal Response**
> > >
> > > Very glad we've answered most of your questions. Thank you very much for your further comments. We will further revise the paper based on your suggestions in future editions, as follows:
> > >
> > > - Indeed, the Lipschitz continuity alone does not guarantee that the integral is bounded. What we wanted to say last time is that we have assumed that the function is Lipschitz continuous, so as long as the domain of the integral, that is, $\mathcal{S}$ and $\mathcal{A}$ are bounded, the result of the integral is bounded. But in the last version, due to time constraints, we missed the further assumption that $\mathcal{S}$ and $\mathcal{A}$ are bounded. Thank you so much for reading through and spotting this question. We will add the relevant assumption that $\mathcal{S}$ and $\mathcal{A}$ are bounded in the future edition. Of course, this assumption does not hold for some environments, but this assumption is only used for theoretical construction and will not affect the performance of our algorithm.
> > >
> > > - Thank you very much for your good and rigorous suggestion. We've thought carefully about your suggestion, as well as that of Reviewer y8C2. We feel that your suggestions do not conflict, and make this paper more rigorous in two ways. Reviewer y8C2 means that for an environment where the set of cycles have measure zero, we can appropriately relax the assumption to almost-sure to make it more general. What you're suggesting is that in many reinforcement learning environments, the set of cycles would not have measure zero, hence strict assumption is required for these environments. We will incorporate your comments into revisions to future editions. And try to make the wording tighter and the assumptions more granular. In particular, "almost-sure" scaling will only be used in a "zero measurement" environment. For other environments, we still make strict assumptions.
> > >
> > > - As for the sentence "This assumption is reasonable because for many continuous environments it is difficult to form cycles in trajectory without special constraints."
> > > We will modify it to:
> > > "This assumption is directly reasonable for some environments, such as when the measure of the set of cycles is zero (i.e., $\mu(\\{s_0,...,s_t\\})=0$), it is difficult to form cycles in a continuous space. As for other tasks, especially those that are prone to form cycles, we can add time steps in the state space to satisfy this assumption."
> > >
> > > Thank you again for your careful review, which has made a great contribution to the improvement of this paper. We are also very happy to receive your positive comments on this paper. Best regards.

---

> > > > ### Comment · Reviewer_Y6ub · 2022-12-12
> > > > **Final response**
> > > >
> > > > I apologize for the late response. All those last points are clear and make perfect sense thank you!
> > > >
> > > > I'm happy to increase my score.

---

> > > ### Author Response · Authors · 2022-12-08
> > > **Response 2 to Post-rebuttal Response**
> > >
> > > Dear Reviewer,
> > >
> > > The rebuttal window is closing soon, so please feel free to let us know if you still have questions. We sincerely hope to increase the exposure of this paper in the community and make contributions to the community. Thank you very much.
> > >
> > > Best regards.

---

> ### Author Response · Authors · 2022-11-26
> **Response to Reviewer Y6ub**
>
> Dear reviewer Y6ub,
>
> We greatly appreciate your valuable suggestions, which have helped us improve the quality of the paper significantly. Where possible, we have added some assumptions to make the definitions related to integral and acyclic in the mathematical model more rigorous, and we discussed the necessity and rationality of the assumptions in the appendix. Our work is the first experimental validation of continuous GFlowNets. Please let us know if you have other questions or comments.
>
> Since the discussion window between reviewers and authors is ending, we sincerely look forward to your reevaluation of our work and would very appreciate it if you could raise your score to boost our chance of more exposure to the community. Thanks a lot!

---

### Official Review · Reviewer_y8C2 · 2022-10-24

**Confidence:** 5
**Correctness:** 3
**Technical Novelty And Significance:** 3
**Empirical Novelty And Significance:** 3
**Recommendation:** 8

**Clarity, Quality, Novelty And Reproducibility:**

Quality, clarity: Good. No major writing issues; the exposition will be easy to follow for a reader who is familiar with GFlowNet basics.

Originality: Very good. However, there could be a better discussion of the relationship to other recent work on GFlowNets, specifically, possible limitations in comparison with DB [Bengio et al., 2022b] and TB [Malkin et al., 2022] objectives.

Reproducibility: Good. Simple and illustrative code is provided.

**Strength And Weaknesses:**

Strengths:
- (+++) This is the first paper to consider GFlowNets with continuous action spaces, which is an important methodological step.
- (+) It is also the first time that GFlowNets are applied to reinforcement learning environments (discrete or continuous).
- (+) Clear and (mostly) theoretically sound exposition.
  - But see my suggestions / comments below.
- (+) **Update, 10 November:** A strength that I neglected to point out in the initial review is that this paper considers environments where inversion of actions cannot be done analytically -- the "transaction network" has to be learned to find the parent (state,action) pairs. This is an interesting idea and the first time such a method is combined with GFlowNets.

Weaknesses / questions:
- (--) It is hard to interpret and contextualize the experiments, at least for a reader who is closely familiar with GFlowNets but not with these particular RL environments. It would be very helpful to have some figures showing some sampled trajectories in each environment.
  - Even better would be a demonstration on a toy environment, like a continuous generalization of the 2D hypergrid from [Bengio et al., 2021a].
- (-) The proposed approach to GFlowNets with continuous action spaces has some weaknesses compared with alternatives. (Ideally, these alternatives would be compared with in the experiments, but at least they should be discussed in the text.)
  - [Bengio et al., 2021b] writes that the detailed balance (DB) objective, which requires estimates of a state flow and parametric forward and backward policies, can be applied to continuous action spaces: the policy likelihoods in this objective is simply replaced by probability densities. The trajectory balance (TB) objective from [Malkin et al., 2022] can be applied in this setting as well.
  - DB and TB have an advantage over flow matching in that they do not require evaluating the flow model on multiple parents of a state, and this advantage becomes more prominent in terms of computation cost when the number of parents is large (for discrete spaces). In the setting of this paper, the computation cost grows with the number of samples used for the estimation of inflow, and thus DB or TB could have a strong computational advantage, in addition to the known benefits of TB for faster convergence.
  - The issue of the variance of the in- and out-flow estimators should receive more attention. (Is the bound involving the Lipschitz constant practically useful?) Also, it should be mentioned that even if the the in-flow and out-flow estimators are unbiased, the estimators of **log**-in- and out-flows, which are used in the loss of equation (20), are not unbiased.
- (-) On the mathematical formulation: The definition of child set on p.3 uses the trajectory history, which makes it impossible for the Markovian condition on the density of the forward policy to hold (though in many settings it holds almost everywhere).
  - In addition, this definition looks tailored to the case when the sets of states reachable in any given number of steps are open sets in a fixed "universe", which is natural for modeling motion in space, but is not very general. For instance, it does not make sense to talk about new states equalling past states in settings where the dimensionality of the reachable state space grows with the number of steps, such as the obvious continuous generalization of [Zhang et al., 2022b].
  - I wonder if a better approach to acyclicity would be to remove the constraint in the definition of child set and instead to say something like: "in our experiment domains, **the set of trajectories in which states repeat has zero measure under the induced trajectory flow** if all policies are absolutely continuous with respect to the standard measure on A, so such trajectories do not affect the analysis", similar to current Remark 2.
  - On a related note, I did not find in the text the answer to whether the time step (number of actions from the initial state) was used as an input to the policy model. If the time step is part of the state, then acyclicity is guaranteed. It does not change the math in this case, but could be necessary in RL problems with discrete action spaces, where not appending the time step would result in cycles having nonzero measure.
  - (A **very** simple way to deal with the cycle issue without having to talk about zero-measure sets: say that the state includes the information about the time step, so cycles are impossible, but the policy ignores the time step part of the state.)
- (-) Related work could use some improvements:
  - "In Malkin et al. (2022), the trajectory balance loss is proposed for GFlowNets to explore the capabilities of previously used objectives" -- this isn't accurate. In fact, that paper proposed and tested a new objective (TB) and tested another objective that had not been used before (DB).
  - "In Bengio et al. (2021b), an idea based on hybrid state is presented to make GFlowNets suitable for continuous tasks" -- this is also inaccurate (see the comment on DB and TB for continuous spaces above -- the proposal in that paper does not require hybrid states).
  - The differences between CFlowNets and continuous SAC (pp.7-8) are hard to understand and could be reworded. To my understanding, (1) means that a Gaussian policy in SAC is less expressive than one that uses a general unnormalized action p.d.f. F(s,a), and (2) means that SAC wants each trajectory leading to a state to have the same likelihood (which should be proportional to the reward), while CFlowNets want the sum of likelihoods of trajectories leading to a state to be proportional to its reward.

Questions and minor comments:
- How were hyperparameters chosen for CFlowNets and baselines?
- Equation (3) seems to have a bug. As written, the integral on the right is usually 0, since it is taken over both $s$ and $a$, but restricted to a lower-dimensional subspace of ${\cal S}\times{\cal A}$. I think it should instead be this: $\int_s\left(\sum_{a:T(s,a)=s_t}F(s,a)\right) ds$.
  - Should there also be a compactness or similar assumption on $A$, so that any function on it defined by a neural net is integrable (and, less importantly, has finite Lipschitz constant)?
- Please use curve markers or line styles, and not just colours, to distinguish algorithms in Figure 3.
- Some of the citations to GFlowNet work are incomplete.
  - [Bengio et al., 2021a] was published in NeurIPS 2021.
  - [Zhang et al., 2022b] was published in ICML 2022.
  - [Deleu et al., 2022] was published in UAI 2022.
  - [Malkin et al., 2022] will be published in NeurIPS 2022.

**Summary Of The Paper:**

This paper proposes a generalization of GFlowNets to environments with continuous action spaces. The continuous version of the flow matching condition, which involves integrals rather than sums, is approximately enforced by using a Monte Carlo estimator of the in- and out-flows. Experiments are performed on three continuous control domains, where the proposed algorithms show strong performance.

**Summary Of The Review:**

My main reason for leaning positive for this paper is the novelty of the problem studied in two respects: using GFlowNets for RL tasks to encourage exploration and providing the first experimental validation of continuous-time GFlowNets, albeit with a different approach that what has been suggested -- but not empirically tested -- in past work. ~There are some issues with the mathematical foundations, but I believe they are fixable by introducing some mild assumptions.~ Post-rebuttal update: The response have adequately addressed my questions and concerns. I would like to see this paper published because it contributes several important GFlowNet firsts (continuous action spaces, application to RL problems, learnable inversion of actions).

---

> ### Author Response · Authors · 2022-11-19
> **Response to Reviewer y8C2 (Part 1)**
>
> Dear Reviewer y8C2,
>
> Thank you for your comments. We will make the modifications to the paper for a better understanding based on your suggestions.
>
> [**Question 1:**] *About the visualization of the experiment environments.*
>
> [**Response for Comment 1:**] Thanks for your suggestion, we have added the sampled trajectories of the Point-Robot-Sparse tasks, as well as the motion visualization of the Reacher-Goal-Sparse task and Swimmer-Sparse task in appendix D, to help with better understanding. For the Reacher-Goal-Sparse and Swimmer-Sparse tasks, we failed to add sampled trajectories, because these MuJoCo environments are difficult to visualize trajectories.
>
> [**Question 2:**] *About the detailed balance (DB) and trajectory balance (TB).*
>
> [**Response for Comment 2:**]
> Thank you very much for your suggestions. We had updated the related works as the following:
> [Bengio et al., 2021b] and [Malkin et al., 2022] propose detailed balance (DB) and trajectory balance (TB) objectives, which use parametric forward and backward policies in the objective function. These new objective functions do not require evaluating the flow model on multiple parents of a state, which is more efficient, especially for high-dimensional environments. [Bengio et al., 2021b] and [Malkin et al., 2022] mentioned that these objective functions can also be used in continuous scenarios by replacing the policy likelihoods in the objective with probability densities. A possible disadvantage is that it is not easy to estimate $P_F$ and $P_B$ in a continuous environment, since the state space is much larger than in a discrete scenario, and a small error in modeling probability densities can greatly affect the final performance. How to combine DB and TB with CFlowNets will be a worthy future work.
> As for the experimental of DB and TB for continuous tasks, we have only completed a simple attempt so far, but it is a pity that we have not achieved a good convergence effect. We try to use neural networks to learn $P_F$ and $P_B$, but it seems that this is not easy to learn in continuous tasks. In addition, due to time constraints, our failure to optimize the algorithm to the best is also a major reason. Therefore, we can only leave this work to future work, which is discussed in the paper.
>
> [**Question 3:**] *About the variance of the in-flow and out-flow.*
>
> [**Response for Comment 3:**] Many thanks for your question. As for the variance of the inflow and outflow estimators, we added a simulation to show that the Lipschitz constant is a finite constant, please see Appendix A in the main paper. Lipschitz continuous is a common assumption of neural networks, just some quick examples: [1]-[4] all use this assumption to prove the convergence of algorithms. As for the log-in- and log-out- flows, they indeed are not unbiased. Although we cannot prove the unbiased property of the log-in- and log-out- flows, simulations show this approximation works well. We highlight this in the main paper. Many thanks for your carefully review, this is indeed a place that is easy to overlook.
>
> [1] Du, Simon, et.al., . "Gradient descent finds global minima of deep neural networks." In International conference on machine learning, pp. 1675-1685. PMLR, 2019.
>
> [2] Jacot, Arthur, et.al., "Neural tangent kernel: Convergence and generalization in neural networks." Advances in neural information processing systems 31 (2018).
>
> [3] Allen-Zhu, et.al., "A convergence theory for deep learning via over-parameterization." In International Conference on Machine Learning, pp. 242-252. PMLR, 2019.
>
> [4] Alistarh, Dan, et.al., "The convergence of sparsified gradient methods." Advances in Neural Information Processing Systems 31 (2018).
>
> [**Question 4:**] *About the acyclicity.*
>
> [**Response for Comment 4:**] We greatly appreciate your constructive comments. We removed the constraints in these definitions following your suggestion and added an assumption to guarantee acyclicity. The corresponding necessity and rationality are discussed in Appendix A.
>
> Also, indeed as you stated, we can add time steps directly in the state space to satisfy the assumptions. We highlighted this in our main paper. For our current simulation, we did not add the time step to state. Since in our task it is difficult to generate cycles due to the fact that "state repetitions have zero measure". The performance is the same whether we add time steps or not.
>
> [**Question 5:**] *About the related work.*
>
> [**Response for Comment 5:**] We greatly appreciate your good suggestions.  The improvements you mentioned about related works have all been revised.

---

> > ### Author Response · Authors · 2022-11-19
> > **Response to Reviewer y8C2 (Part 2)**
> >
> > [**Question 6:**] *About hyperparameters.*
> >
> > [**Response for Comment 6:**] We have added the relevant description in the appendix D.5. Our hyperparameters are set as follows:
> >
> > *"As for "Total Timesteps", "Start Traning Timestep", "Max Episode Length", "Actor Network Hidden Layers", "Critic Network Hidden Layers", "Optimizer", "Learning Rate", and "Discount Factor", we set them the same for all algorithms for a fair comparison. As for these specific parameters for baseline algorithms, we remain them the same as those in the original code to achieve good performance. As for these specific parameters of our CFlowNets, we set the number of sample flows to 100 and the action probability buffer size to 1000 to tradeoff the performance and computational load. Note that CFlowNets dose not require as large a replay buffer size as other RL algorithms, since the exploration ability of CFlowNets is better than that of others. And a good policy can already be learned from a small replay buffer. This is also an advantage of CFlowNets compared to RL based algorithms. "*
> >
> > [**Question 7:**] *About bugs of Equation (3).*
> >
> > [**Response for Comment 7:**] Thanks a lot for pointing this out. We added an assumption to make this equation more strict. In addition, we discuss the necessity and rationality of the assumptions in the appendix. And re-revised the relevant proofs.
> >
> > [**Question 8:**] *About curve markers.*
> >
> > [**Response for Comment 8:**] Curve markers in Figure 3 are updated.
> >
> > [**Question 9:**] *About citations.*
> >
> > [**Response for Comment 9:**] Missing citations are added.

---

> ### Author Response · Authors · 2022-11-26
> **Response to Reviewer y8C2**
>
> Dear reviewer y8C2,
>
> We very much appreciate your valuable suggestions, which have helped us significantly improve the quality of the paper. Where possible, we have added a discussion of the DB and TB algorithms and made significant modifications to the mathematical model. Our work is the first experimental verification of continuous GFlowNets. Please let us know if you have other questions or comments.
>
> Since the discussion window between reviewers and authors is ending, we sincerely look forward to your reevaluation of our work and would very appreciate it if you could raise your score to boost our chance of more exposure to the community. Thanks a lot!

---

> > ### Comment · Reviewer_y8C2 · 2022-11-28
> > **Response to response**
> >
> > Thank you for the answers to my questions. The changes have mostly addressed my concerns and I have no further questions, although I do wonder whether Assumption 1 could be relaxed to almost-sure acyclicity, rather than strict acyclicity. The sentence that Assumption 1 "is reasonable because for many continuous environments it is difficult to form cycles" seems to be saying exactly that in "typical" environments, cycles have zero measure.
> >
> > On a minor note, the updates to the paper seem to have introduced many little errors/typos that should be corrected in a final or revised version. A few, starting from the top:
> > - "GFlowNet aims to generate distribution proportional to the rewards" (Do you mean "generate samples from a distribution"?)
> > - "define a distribution proportional to the rewards over terminating states, i.e., the parent states of the final states" (Do you mean "final **state**" $s_f$ as in [Bengio et al., 2021b], using the convention that there is a single sink state and its parents are the terminating states?)
> > - "important sampling" -> "importance sampling"

---

> > > ### Author Response · Authors · 2022-11-28
> > > **Response to further questions**
> > >
> > > Dear reviewer y8C2,
> > >
> > > We are very happy to have answered most of your questions. Thank you very much for these further good suggestions.
> > >
> > > Indeed, unless there are some special environments (such as Pendulum-with-Wall in Figure 5), it is true that other environments are almost-sure acyclic. We will adjust Assumption 1 to make it more general. This is indeed a very good suggestion!
> > >
> > > Many thanks again for reading through and catching these typos. We will continue to carefully revise the paper until the final version. As for these typos, we will fix them.
> > > - Yes, we will modify that into "generate samples from a distribution".
> > > - Yes, we will add the details and the reference [Bengio et al., 2021b].
> > > - We will modify that into "importance sampling".
> > >
> > > We sincerely appreciate your great contributions to improve the quality of this paper, and believe that these good suggestions will help GFlowNets to be used in a wider range of scenarios to better help the community.

---

> > > > ### Comment · Reviewer_y8C2 · 2022-12-06
> > > > **Wrong / inconsistent notation + code question**
> > > >
> > > > I have one more question about the paper and a related question about the code.
> > > >
> > > > First, I'd like to point out an error on page 4, specifically the paragraph after equation (14) and the following equation. It is written that the reward is 0 for all states different than $s_f$ (the unique final state of the DAG). However, it does not make sense to talk about $R(s_f)$, since the reward depends on the state from which we terminate, not on the unique final state. Should it be $R(s\rightarrow s_f)$ (the reward received when terminating from $s$), or, in a different convention where all states that have an edge to $s_f$ have no other children, $R(s_{T-1})$ where $s_T=s_f$ is the final state in the trajectory?
> > > >
> > > > (Similarly, when you write $R(s_t)$ in the equation below (14), you seem to mean the reward when transitioning from $s_t$ to $s_{t+1}$. The last term in that equation involves children of $s_f$, which does not make sense. Should the sum go only up to $T-1$?)
> > > >
> > > > Second, I have some confusion about how intermediate rewards are treated in the code. In the grid and reacher environments, the control cost is added to the out-flow at intermediate states, but in the swimmer environment, the intermediate rewards are set to 0 and the **control costs are added to the episode reward**.
> > > > - What is the reason for the difference?
> > > > - I think the approach taken for the swimmer violates a key assumption of GFlowNets, that the reward depends on the state from which we transition to $s_f$ and not on the trajectory used to get there. Could you please comment on this?
> > > >
> > > > Thank you for your clarifications.

---

> > > > > ### Author Response · Authors · 2022-12-07
> > > > > **Response to notation + code questions**
> > > > >
> > > > > Thank you very much for your further questions.
> > > > >
> > > > > - Thank you so much for spotting these typos. Indeed, $R(s_f)$ is imprecise and should be modified into "for sparse reward environments, i.e., $R(s) = 0$ except $R(s\rightarrow s_f)$".
> > > > >
> > > > > - As for the loss function below equation 14, earlier we introduced indicator functions to avoid computing the children of the final state. But another reviewer suggested that we remove the indicator function. Because for a state without child nodes and rewards, its outflow and reward are defined as 0, which does not affect the result. According to your suggestion, in order to be more rigorous, we will modify the loss to
> > > > > $$
> > > > > {\mathcal{L}} (\tau) =\sum\limits_{s_{t} \in \tau} \Bigg(  ... \Bigg)^{2}
> > > > > $$
> > > > > where $\tau = (s_1,...,s_n)$ and $R(s_n \rightarrow s_f) \neq 0$.
> > > > >
> > > > > As for the code, we followed the assumptions of the paper and used sparse rewards in all tasks.
> > > > >
> > > > > - a) In the grid task, rewards are issued by measuring the distance between the agent’s current position and the target node, where the intermediate rewards are set to 0 in the step function of the environment (line 35 of point\_env.py).
> > > > >
> > > > > - b) In the reacher task, rewards are issued by measuring the distance between the agent’s current position and the randomly generated target, where the intermediate rewards are set to 0 through the wrapper (line 59 of cfn\_reacher.py).
> > > > >
> > > > > - c) In the swimmer task, rewards are issued by measuring the distance between the agent’s current x-coordinate position and the x-coordinate position at the last moment, where the intermediate rewards are set to 0 when appended in the buffer (line 286 of cfn\_swimmer.py).
> > > > > What you may see is the code that uses episode rewards to calculate the distance between the final x-coordinate position and the position of the starting point(line 282 of cfn\_swimmer.py), and then by adding the absolute function to the episode reward to indicate the greater the distance, the greater the reward value (line 293 of cfn\_swimmer.py).
> > > > > We have re-adjusted the way the code is written to avoid unnecessary misunderstandings, and our previous code can be split into:
> > > > > ```
> > > > > class RewardShapeWrapper(gym.Wrapper):
> > > > >     def __init__(self, env):
> > > > >         super().__init__(env)
> > > > >         self.x = 0
> > > > >
> > > > >     def reset(self, **kwargs):
> > > > >         self.x = 0
> > > > >         return self.env.reset(**kwargs)
> > > > >
> > > > >     def step(self, action):
> > > > >         observation, reward, done, info = self.env.step(action)
> > > > >         position_x = info['x_velocity']
> > > > >         self.x += position_x
> > > > >         if done:
> > > > >             return observation, math.fabs(self.x), done, info
> > > > >         else:
> > > > >             return observation, 0.0, done, info
> > > > > ```
> > > > > The episode reward of the above three environments depends on the state from which we transition to $s_f$.
> > > > >
> > > > > Thank you very much for your suggestions and questions, which are of great help to the improvement of our paper.

---

> > > > > > ### Comment · Reviewer_y8C2 · 2022-12-07
> > > > > > **Thank you**
> > > > > >
> > > > > > Thank you for clarifying! I looked closely at the code because I was trying to understand the equation, but neglected to consider the logic in RewardShapeWrapper. Apologies for the confusion and for the somewhat last-minute question.

---

### Decision · Program_Chairs · 2023-01-20

**Decision:**

Accept: poster

**Justification For Why Not Higher Score:**

A spotlight could be considered but is not necessary for this paper.

**Justification For Why Not Lower Score:**

Three reviewers that are very knowledgable on the topic championed for acceptance.

**Metareview: Summary, Strengths And Weaknesses:**

This paper proposes a continuous formulation of GFlowNet, for both the action and state spaces, by converting those spaces into continuous spaces and converting the sums of flow-matching into integrals. While there were initial concerns by the reviewers, the authors have done a great job at addressing them with many reviewers increasing the score. The key strength of this paper is not only the extension of GFlowNets to both continuous action and state spaces, but also the empirical validation that this works in encouraging experiments. I recommend accepting this paper.

**Note From Pc:**

if the above contains the word "oral" or "spotlight" please see: "oral" presentation means -> notable-top-5% and "spotlight" means -> notable-top-25%. As stated in our emails, we are disassociating presentation type from AC recommendations